# Seismic performance of base-isolated structures for swimming pool reactors under different foundation conditions

Jie Zhao[1]*, Jianshan Wang[1], Jiehua Huang[2]

1 School of Architectural Engineering, Dalian University, Dalian, Liaoning, China, 2 Business Office, Guangdong Shunkong Development Co., Ltd. Ronggui Branch, Foshan, Guangdong, China

* zhaojie_gd@163.com

**Data Availability Statement:** All relevant data are within the paper and its Supporting Information files.

**Funding:** National Natural Science Foundation of China (authorization number: 52108437),

## Abstract

The swimming pool reactor (SPR) is an innovative and environmentally friendly heating source. An SPR building was selected as the research subject, and a 3D dynamic interaction model incorporating the liquid sloshing effect was created using ANSYS software and the secondary development characteristics of user-programmable features (UPFs). Energy dissipation from scattered waves was accounted for using viscous-spring boundary elements, while the dynamic hydraulic effect was modeled via the Housner equivalent mechanical model. Considering soil-structure interaction (SSI) effects, this study examines the impact of isolation measures on the structure's seismic mitigation performance. It investigates how varying foundation conditions affect the seismic resistance of the isolated structure. Results indicate that seismic isolation ratios for acceleration, floor response spectra, displacement, and base shear diminish as site stiffness decreases. However, regarding sloshing wave height, seismic isolation amplified the height under all conditions but remained within safe limits. These findings offer valuable insights for seismic design across different SPRs.

## Introduction

In northern China, winter heating has always been a top priority. Compared to traditional fossil fuel heating, nuclear heating provides the benefits of being clean, renewable, and cost-effective. Moreover, with China aiming for dual carbon goals—achieving carbon peaking and carbon neutrality—the nation has pursued nuclear heating research vigorously. In this context, the swimming pool reactor has gained prominence. The potential applications of the swimming pool reactor are promising, positioning it to become a widely accepted, market-leading solution that will deliver warmth to the people of China. Applying seismic isolation technology in swimming pool reactors provides new technical methods for earthquake disaster mitigation and introduces innovative research, development, and promotion concepts.

The "Belt and Road" strategy has facilitated the gradual inland expansion of nuclear power plant construction. However, the scarcity of ideal bedrock sites has raised concerns over site

sponsored by Xunqiang Yin, and the funding amount is 30000 RMB. The funders had no role in study design, data collection and analysis, decision to publish, or preparation of the manuscript. Jie Zhao received funding from the sponsor Xunqiang Yin.

**Competing interests:** The authors have declared that no competing interests exist.

adaptability to soil conditions, becoming a key challenge for future nuclear power plant construction. Seismic isolation technology has effectively expanded the range of suitable nuclear power plant sites by significantly reducing the superstructure's floor response spectra during seismic activity. Extensive research has been conducted globally on this topic. Through shaking table experiments, it was found that base isolation can significantly reduce the hydrodynamic pressure on storage tanks [1]. To simulate dynamic response tests, some researchers applied laminated lead rubber bearings on the tanks [2], and the results indicated that using isolation bearings can significantly reduce the dynamic response of the tanks. For the structure of the fourth-generation nuclear power plant, the researchers carried out the seismic response analysis of the three-dimensional seismic isolation of the whole foundation [3], and carried out the dynamic characteristics under the design reference ground motion, the floor response spectrum analysis and the characteristics analysis of the isolation layer under the overdesign reference ground motion through the design of the isolation layer and the comparison of the models before and after the isolation. Additionally, based on the principle of curved motion and the characteristics of pre-compressed spring expansion and contraction, a negative stiffness damping system was proposed [4], which can effectively reduce both the upper acceleration response of the nuclear power plant and the displacement response of the isolation layer. In the nuclear island building, a refined finite element model and SAP2000 software were used for analysis, proposing overall isolation design objectives for the nuclear island [5], which can effectively improve the adaptability of the nuclear power plant site. Through dynamic time history analysis [6], the dynamic responses of structures with and without isolation under seismic actions were studied, comparing the effects of base isolation on structural acceleration and displacement responses.

The seismic performance of three types of combined variable stiffness seismic isolation systems in the nuclear island structure was also discussed [7], and the engineering design and research of the basement seismic isolation technology were carried out in the context of a nuclear safety-related plant project of "Hualong One" [8], and the results showed that the seismic isolation technology could effectively reduce the horizontal seismic response of the structure and its ancillary equipment to improve the seismic safety margin. In addition, a new horizontal combined seismic isolation scheme [9] is proposed, which can avoid the tension of the isolation bearing and the overturning of the structure, and compares with the results of the seismic isolation scheme and the foundation isolation scheme.

In order to achieve a small displacement of the isolation layer in the static load stage and a good damping effect in the dynamic load stage, a three-dimensional isolation system with high static and low dynamic vibration was designed, which consists of a horizontal isolation unit and a high-static and low-dynamic isolation system [10,11] (composed of oblique rubber bearings and negative stiffness devices). In addition, a hybrid control system with additional lateral viscous damping for the three-dimensional seismic isolation structure of semi-embedded/fully embedded small stacks [12] is proposed to control the horizontal acceleration, horizontal displacement and sway response at the same time, without affecting the vertical seismic isolation effect.

In the impact of seismic response on nuclear power structures, the structural-foundation dynamic interaction (SSI effect) cannot be ignored. Considering the influence of structural-foundation dynamic interaction (SSI effect) on nuclear power structures [13], a sensitivity analysis of foundation parameters was conducted to explore the uncertainty of foundation rock and soil parameters at different sites on the floor response spectra of the nuclear island building. To study the impact of the SSI effect on nuclear power structures [14], the safety shell structure of the third-generation nuclear reactor AP1000 was selected as the research object, and a 1/40 scale model of the safety shell was created for shaking table tests, using a flexible soil box to eliminate boundary effects.

In numerical model analysis, there are always two types of uncertainties: aleatory uncertainty (i.e., materials, geometric properties and actions) and epistemic uncertainty (i.e., modeling). Aleatory uncertainty arises from the inherent randomness in data or models. For example, there may be measurement noise in the experimental data, or there may be random disturbances in the simulation. Epistemic uncertainty arises from incomplete knowledge of the model or system, including simplification, approximation, or inaccuracy of assumptions. For example, simplification of physical models, roughness of computational grids, etc. By improving the model, increasing computational accuracy, and reducing assumptions, epistemic uncertainty can be reduced. A novel reliability analysis method is proposed for multi-objective systems that contain both aleatory uncertainty and epistemic uncertainty [15]. This method combines probability and evidence models, and successfully solves the failure probability of this structure by constructing a second-order limit state equation and corresponding reliability analysis method [16]. Overview and comparison of three different methods for estimating global safety factors: Probability Format (PF), Partial Safety Factor Format (PSFF), and Global Resistance Format (GRF) in the Global Resistance Method (GRM). The purpose is to address the effectiveness of different methods for reliability assessment of RC components in GRM, as well as the correlation between aleatory and epistemic uncertainties [17,18].

Considering the specific conditions of the swimming pool reactor structure and relevant industry standards, This paper designs a seismic isolation structure and incorporates a base isolation layer. Considering SSI effects, this study explores the impact of isolation measures on the structure's seismic mitigation performance. It examines how varying foundation conditions influence the seismic resistance of the isolation structure.

## Principles of computational methods

### Viscous-spring artificial boundary

**Three-dimensional viscous-spring artificial boundary.** In dynamic response studies involving soil structure interaction, such as in caverns, nuclear power plants, and immersed tunnels, the viscous-spring artificial boundary is pivotal in transforming the semi-infinite domain problem into a finite model analysis of the structure and near-field foundation. The viscous-spring artificial boundary static-dynamic combined model (Fig 1). This model considers the soil-structure dynamic interaction by creating non-thickness compact support viscous-spring artificial boundary elements. The foundation part can consider the foundation's heterogeneity, irregular geometric shape, and the effects of various static loads.

The viscous-spring boundary has evolved from the viscous boundary and is a continuous local boundary model widely applied in addressing SSI effects. After a lot of research and theoretical derivation, a highly recognized three-dimensional viscoelastic boundary was proposed, and on this basis, the equivalent consistent viscoelastic boundary elements were successively developed and promoted [19,20]. In the 3D viscous-spring artificial boundary model, the parameters for each physical component distributed continuously at the foundation boundary nodes (springs with rigid recovery capabilities and dampers with energy absorption characteristics) can be determined using Formulas (1) to (4) to calculate the tangential and normal damping coefficients and spring stiffness coefficients.

$$C_{\mathrm{T}} = \rho \cdot c_{\mathrm{s}} \cdot \Delta A_i \qquad (1)$$

$$C_{\mathrm{N}} = \rho \cdot c_{\mathrm{p}} \cdot \Delta A_i \qquad (2)$$

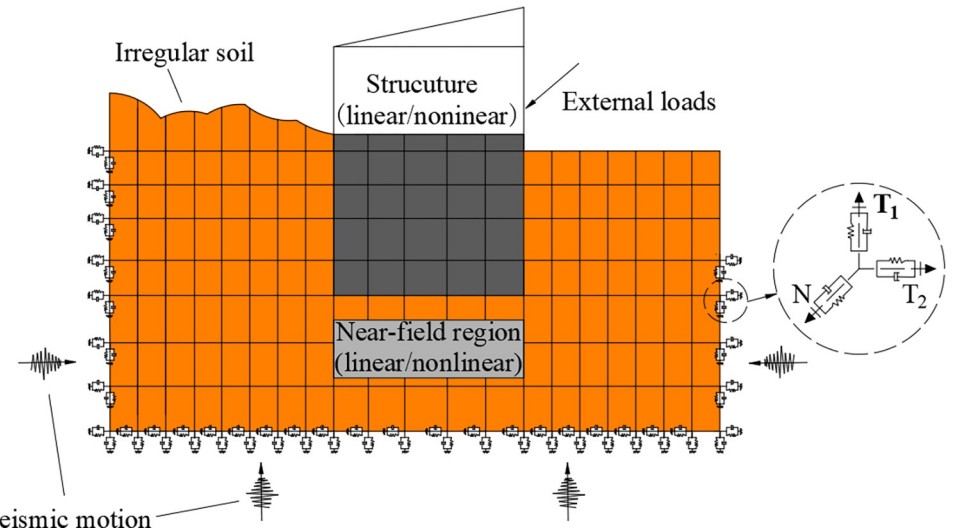

**Fig 1. Schematic diagram of static and dynamic model of viscous-spring artificial boundary.**

$$K_{\mathrm{T}} = \alpha_{\mathrm{T}} \cdot \Delta A_i \cdot G/r \tag{3}$$

$$K_{\mathrm{N}} = \alpha_{\mathrm{N}} \cdot \Delta A_i \cdot G/r \tag{4}$$

In the formulas, $\alpha_{\mathrm{T}}$ and $\alpha_{\mathrm{N}}$ are the tangential and normal spring stiffness correction factors, optimized by free-field analysis $\alpha_{\mathrm{T}}$ with = 3, $\alpha_{\mathrm{N}} = 4$; $c_s$ and $c_p$ represent the propagation rates of seismic transverse and longitudinal waves within the foundation boundary, respectively; $\Delta A_i$ denotes the control area at the node on the viscous-spring artificial boundary; $\rho$ is the density of the infinite continuous medium; $G$ is the dynamic shear modulus of the infinite continuous medium; $r$ is the distance between the secondary scattering field source and the foundation's viscous-spring artificial boundary node.

**Input method for seismic motion.** This paper utilizes the equivalent nodal load input method, which has been widely adopted over the past two decades. The core principle involves converting input seismic motion into equivalent loads on the viscous-spring boundary, treating it as a wave source problem, and facilitating wave motion input following specific mechanical principles. The formula is as follows:

$$F_b = R_b^{ef} + C_b \dot{u}_b^{ef} + K_b u_b^{ef} \tag{5}$$

In the formula, $u_b^{ef}, \dot{u}_b^{ef}, R_b^{ef}$ denote the displacement vector, velocity vector, and corresponding equivalent force vector of the free-wave field at the boundary nodes, respectively, and $K_b$ and $C_b$ are the spring matrix and viscous damping matrix on the boundary, respectively.

**Secondary development of three-dimensional viscous-spring boundary element.** The User Programmable Features (UPFs) is a supplement and expansion of some of the functions of APDL, it is essentially a secondary development at the source code level, through a series of custom files (user subroutines) provided by ANSYS for users to edit and modify, and then use the specified version of the compiler (MS C++ and FORTRAN) to generate a customized version of ANSYS software, so as to achieve the functions expected by users or functions that the software does not have, such as optimization algorithms, definition of user failure guidelines and contact guidelines; development of constitutive models of complex materials, etc.

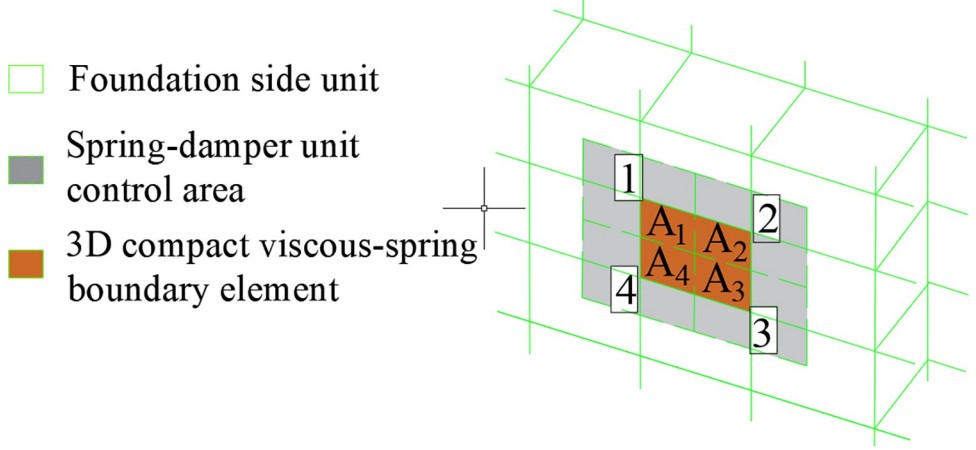

**Fig 2. Customized 3D viscous-spring boundary element.**

The ANSYS program, combined with the implicit integration method and 3D viscous-spring artificial boundary theory, is a platform for batch analysis of the soil-structure interaction model using the parametric design language APDL by leveraging the interactivity between the secondary development tools User Programmable Features (UPFs) and FORTRAN, subprograms written in FORTRAN interface with ANSYS to customize user elements. Interface subprograms based on the element matrix [21] compute all necessary data inputs for new element definitions, including material properties and node coordinates, one by one. This process updates the ANSYS database, simulates the radiation-damping effect of the infinite foundation, provides elastic support to the near-field structure, and implements an equivalent form of seismic motion input. The user element and function implementation source code subprograms are copied to the user folder of the ANSYS installation path, then compiled and linked through the ANS_ADMIN utility program to generate the executable file ANSYS.exe, thereby embedding the 3D viscous-spring artificial boundary element into the element library for quick call-up and real-time simulation. Notably, this element, a custom-defined feature module of the program, is subject to limitations due to the element mesh segmentation, thus requiring a transition using ANSYS's built-in MESH200 element. The implementation principle involves horizontally segmenting the external foundation boundary with MESH200 elements and using APDL control command flow to replace the element type with the user-defined 3D viscous-spring artificial (Fig 2).

**Example verification.** This article is dominated by epistemic uncertainty, and the correctness of viscous-spring boundary is verified through numerical examples to reduce epistemic uncertainty. In this section, a homogeneous elastic half-space model with a rectangular geometry is established. The calculation domain boundary along the length and width directions is set to 40m and 30m, respectively, and 20m along the height direction. The element used is SOLID185, and the mesh size for discretization is 1m×1m×1m, and linear elastic material is assigned to the elements, with specific parameters of ν = 0.25、E = 3.0e7Pa、ρ = 2000kg/m3. A unit impulse displacement wave is vertically input at the bottom of the model, with the formula:

$$u(t) = \begin{cases} 0.5\sin(4\pi t), 0 \leq t \leq 0.5 \\ 0 \qquad\qquad\quad t \geq 0.5 \end{cases} \tag{6}$$

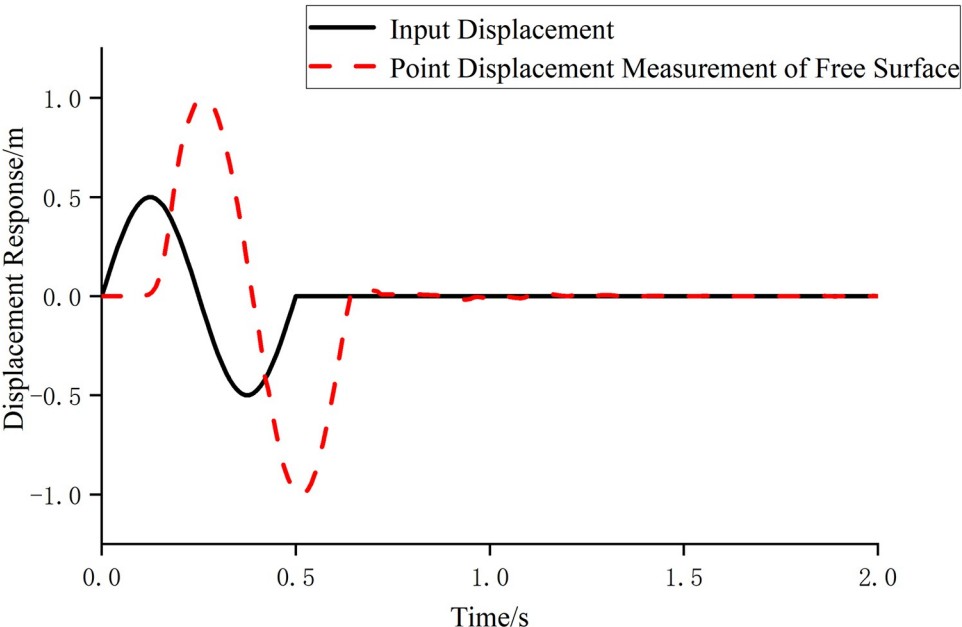

**Fig 3. Comparison of displacement time-history response.**

It shows the displacement time-history response extracted from the free surface of the medium (Fig 3). It is easy to find from the figure that the displacement time history with a doubled amplitude is obtained on the free surface, and it exhibits a specific hysteresis phenomenon compared to the incident displacement time history, showing characteristics of the traveling wave effect. This fully demonstrates that the viscous-spring boundary has a good absorption effect on the wave motion. In summary, the viscous-spring boundary element developed in this paper has considerable accuracy and meets the precision requirements.

## Dynamic hydraulic effects

The reactor pool is the core part of the swimming pool reactor; thus, during an earthquake, the dynamic hydraulic pressure generated by the swaying of the water in the pool is a crucial dynamic load in the seismic analysis of the swimming pool reactor, significantly affecting the stable operation of the reactor pool's heating circuit. ASCE/SEI 4–16 [22] and the Standard for Seismic Design of Nuclear Power Plants [23] recommend using the Housner equivalent mechanical model to simulate the dynamic hydraulic effect for sections that are more regular and non-thin-walled vertical storage containers, such as cylindrical and rectangular containers. The principle of Housner theory is to equivalently divide the liquid in the storage tank from the bottom up into two main parts: one is the impulse mass at the bottom of the tank rigidly connected to the vessel walls, and the other part consists of a series of odd-order sloshing masses elastically connected to the vessel walls at the upper part of the tank (Fig 4). Based on related theoretical deductions and numerical simulations, it is known that the first-order sloshing mass plays a primary role in dynamic analysis. As the order of the sloshing mass increases, the convective pressure of higher-order sloshing masses dramatically decreases, and the vibration attenuation is significant. Hence, this paper only considers and simulates the impact of first-order liquid sloshing. The necessary parameters for the calculation can be determined using the following formulas: Formulas (7)–(11) sequentially calculate the equivalent impulse mass and its effective height, equivalent sloshing mass and its effective height, and spring

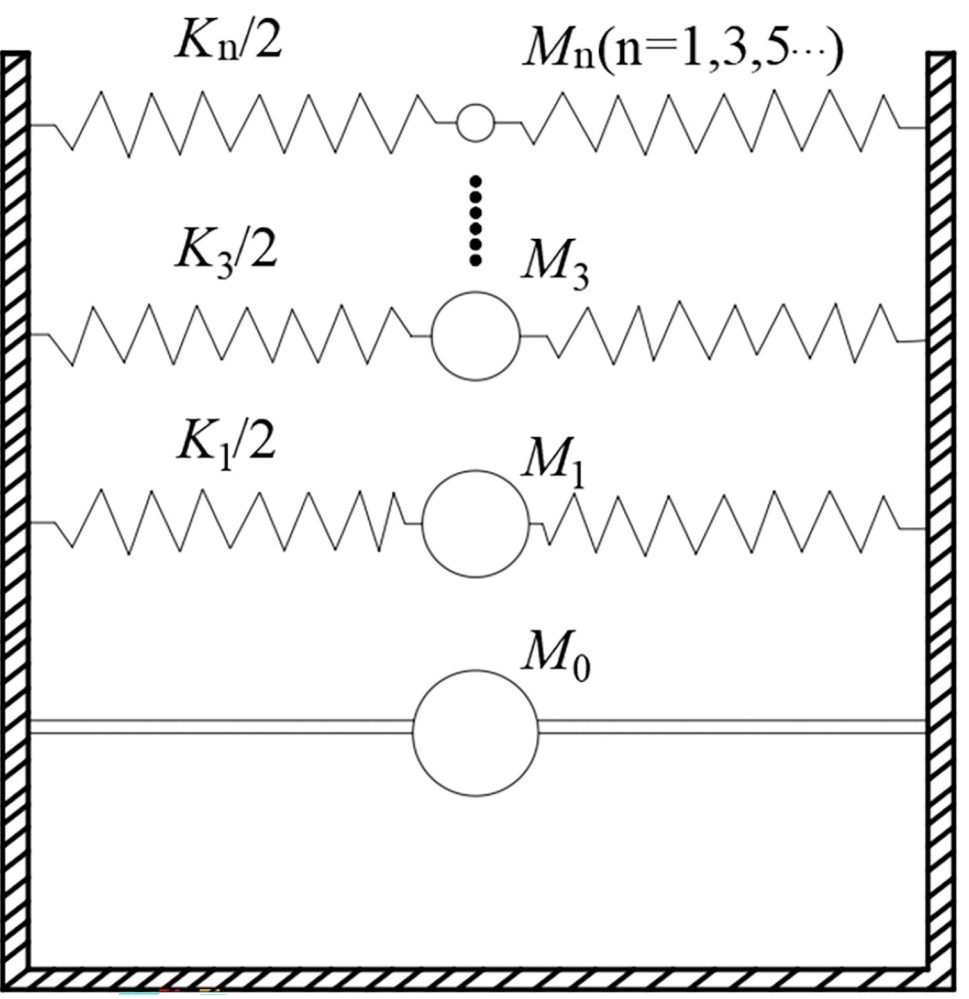

**Fig 4. Schematic diagram of Housner theory.**

stiffness for cylindrical storage containers. Similarly, as a rectangular, its interior deionized water solution used for wet storage of spent fuel can also be simulated using Housner theory, with similar formulas to those for cylindrical storage containers. Detailed calculation formulas are referred to in the literature [24].

For cylindrical liquid storage containers:

$$M_0 = M \frac{\tanh(1.7R/h)}{1.7R/h} \tag{7}$$

$$h_0 = \frac{3}{8}h\left\{1 + \alpha\left[\frac{M}{M_1}\left(\frac{R}{h}\right)^2 - 1\right]\right\} \tag{8}$$

$$M_1 = M \frac{(0.6)\tanh(1.8h/R)}{1.8h/R} \tag{9}$$

$$h_1 = h\left[1 - 0.185\left(\frac{M}{M_1}\right)\left(\frac{R}{h}\right) - 0.56\beta\frac{R}{h}\sqrt{\left(\frac{MR}{3Mh}\right)^2 - 1}\right] \qquad (10)$$

$$k_1 = 5.4\frac{M_1^2}{M}\frac{gh}{R^2} \qquad (11)$$

In the formulas, $R$ is the radius of the container; $M$ and $h$ represent the total mass and depth of the liquid in the container, respectively; and the values for coefficients $\alpha$ and $\beta$ are 1.33 and 2.0, respectively.

## Finite element model

### Overview of the reactor model

A specific model of the swimming pool reactor is embedded within the soil foundation and is classified as an underground structure under Category I seismic item, Its overall schematic diagram (Fig 5). The structure's overall dimensions are 39.15 meters in length, 18 meters in width, and 30 meters in height, with the base elevation at -28 meters. The depths of the deionized water solution in the reactor pool and the spent fuel pool are 24 and 14 meters, respectively. The entire structure of the swimming pool reactor is constructed from concrete, with the top of the reactor pool shielded by a steel plate that is situated above ground level, with a top elevation of +2 meters. The materials employed include concrete of strength grade C40 and steel of grade Q345. Utilizing ANSYS software, a comprehensive model for seismic analysis of the swimming pool reactor structure-foundation system has been established. The SOLID185 element is applied to simulate both the main body of the structure and the

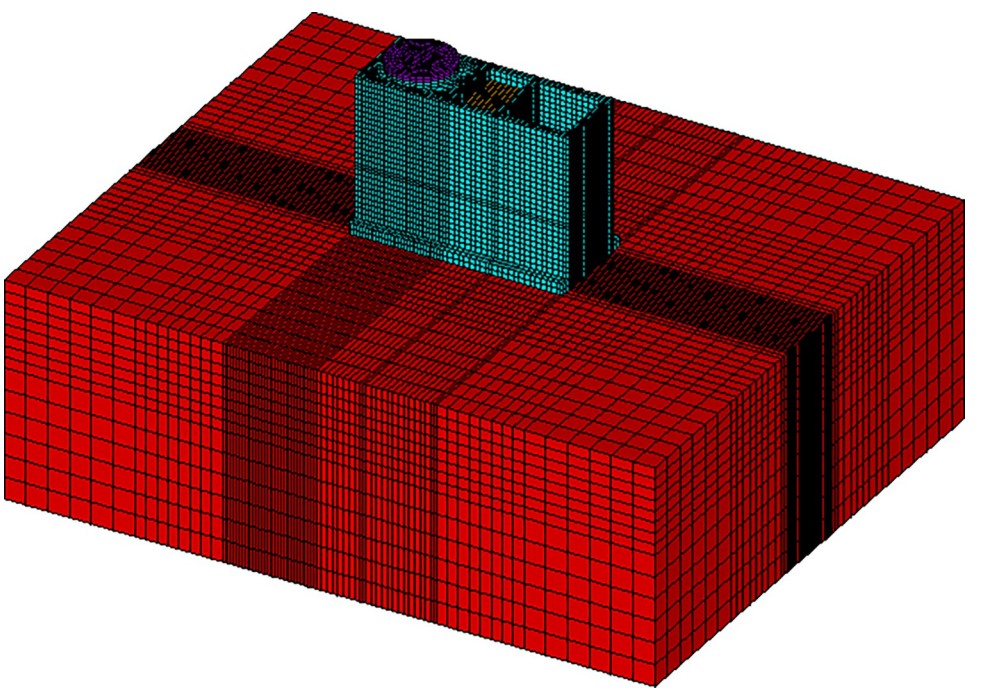

**Fig 5. 3D finite element model.**

foundation. Concurrently, the effects of liquids in the reactor and spent fuel pools are modeled using MASS21 elements and spring elements. For the effects of liquid in the water storage tank and spent fuel pool, FLUID80 elements were used based on the Housner theory, with coupled degrees of freedom used to simulate fluid-structure interaction effects. The finite element simulation's foundation extends 40 meters horizontally and downward from the axis of the swimming pool reactor structure. The foundation was divided into uniform grids as much as possible, with structural element sizes ranging from 0.5 to 2 meters, foundation element sizes shall not exceeding 5 meters. The model consists of 88,031 solid elements, 7,488 fluid elements, and 39 concentrated mass elements, totaling 95,558 elements and 111,171 nodes.

## Selection of calculation parameters and determination of operating conditions

The force convergence criterion for dynamic calculations is 0.05, and the displacement convergence criterion is also 0.05. The soil is modeled using the Mohr-Coulomb model, the concrete structure is considered linear elastic. The paper employs a direct coupling method for fluid and solid, enforcing the equality of degrees of freedom at coincident nodes on the fluid-solid interface to achieve displacement transfer.

Based on Formulas (7) to (11), the parameters for the Housner equivalent mechanical model have been calculated (Table 1). Furthermore, to intuitively demonstrate the sensitivity and regularity of the seismic response of the pool-type reactor under varying site conditions, four types of site conditions are selected for qualitative research. These selections are based on the soil classification from the Code for Seismic Design of Buildings [25] and data from existing or proposed nuclear power plant sites, representing hard rock 1, hard rock 2, soft rock, and soil foundations, respectively. The corresponding operational conditions are defined and analyzed comparatively to illustrate the process of rock-soil transition. The specific calculation parameters and operational condition definitions for the 3D seismic response analysis of the swimming pool reactor structure and foundation soil are detailed in Table 2. According to the Standard for Seismic Design of Nuclear Power Plants [23], the recommended inherent damping factor is 5%

## Determination and implementation of isolation layer parameters

**Selection of elements.** According to the standardized design requirements for structures and nuclear grade equipment and referring to the American nuclear safety structural seismic analysis standards (ASCE/SEI 4–16) [22], this study adopts local plant foundation horizontal isolation technology, temporarily excluding vertical isolation. The model of the base-isolated structure in the present study considers the flexibility of the superstructures and ignores the impact at the base on the adjacent moat wall [26,27]. The isolation bearings are simulated using an equivalent linear mechanical model [28,29], employing COMBIN14 elements in three independent directions to assemble and simulate the seismic isolation bearing (Fig 6).

Table 1. Parameters of Housner equivalent mechanical model.

| Parameter | Reactor pool | Spent fuel pool |
|---|---|---|
| M0/kg | $1.81 \times 10^6$ | $1.14 \times 10^6$ |
| M1/kg | $1.31 \times 10^5$ | $2.98 \times 10^5$ |
| h0/m | 4.51 | 4.04 |
| h1/m | 21.22 | 8.57 |
| k1/N•m-1 | $4.62 \times 10^5$ | $7.59 \times 10^5$ |

**Table 2. Seismic calculation parameters of swimming pool reactor structure.**

| Material | Working Condition | Type of Soil | Shear Wave Speed ($c_s$ /m•s$^{-1}$) | Density ($\rho$ /g•cm$^{-3}$) | Poisson's Ratio ($\mu$) | Elastic Modulus (E/MPa) |
|---|---|---|---|---|---|---|
| Concrete C40 | - | - | - | 2.50 | 0.20 | 42250 |
| Steel Q345 | - | - | - | 7.85 | 0.27 | 207000 |
| Basalt | Condition 1 | Hard Rock 1 | 1684 | 2.35 | 0.29 | 15480 |
| Sandstone | Condition 2 | Hard Rock 2 | 988 | 2.24 | 0.30 | 9210 |
| Sandy Shale | Condition 3 | Soft Rock | 732 | 2.12 | 0.33 | 4360 |
| Silty Clay | Condition 4 | Medium Soft Soil | 212 | 1.89 | 0.48 | 160 |

**Determination of isolation layer parameters.** The total weight of the swimming pool reactor structure is approximately 23,146,750 kilograms. The isolation period for the seismic isolation structure is established based on the seismic acceleration response spectrum at 2.0 seconds, with a damping ratio of 0.3. The introduction of a relatively flexible isolation layer necessitates the determination of its stiffness in various directions, which is critical as it directly influences the seismic safety margin of the structure. The total horizontal stiffness of the isolation layer can be computed using the following formula:

$$K_h = \frac{(2\pi)^2 M}{T^2} \tag{12}$$

In the formula, M represents the total mass of the structure above the isolation layer, and T is the isolation period of the structure. The calculated total horizontal stiffness $K_h$ is $2.28449 \times 10^8$ (N/m).

The horizontal damping of the isolation layer is of the displacement-dependent type [25], with the damping coefficient primarily determined by the damping ratio during shear deformation. This parameter is calculated using the following formula:

$$C_h = 2\xi\sqrt{K_h M} \tag{13}$$

In the formula, $\xi$ is the damping ratio, typically between 20% and 30%, with specific values provided by the manufacturer and varying according to the isolation bearings. The calculated horizontal damping coefficient, $C_h$ is $4.36306 \times 10^7$ (N·s/m).

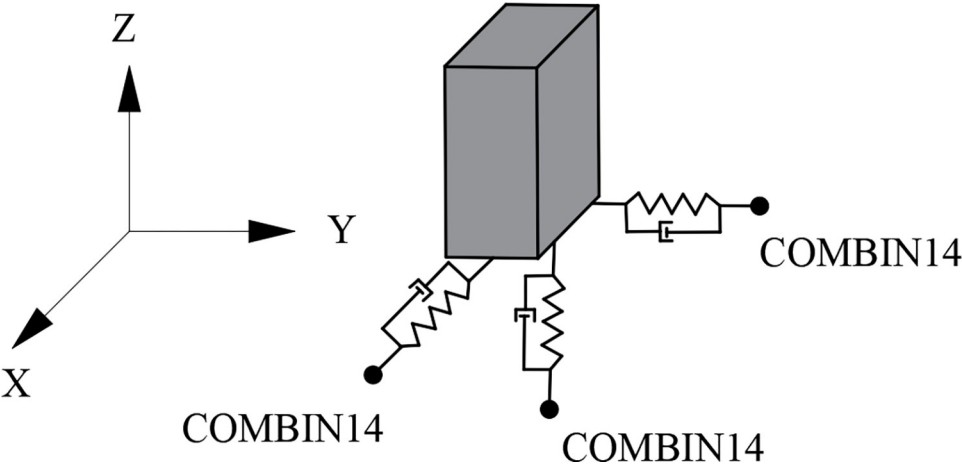

**Fig 6. Schematic diagram of unit combination of isolation bearings.**

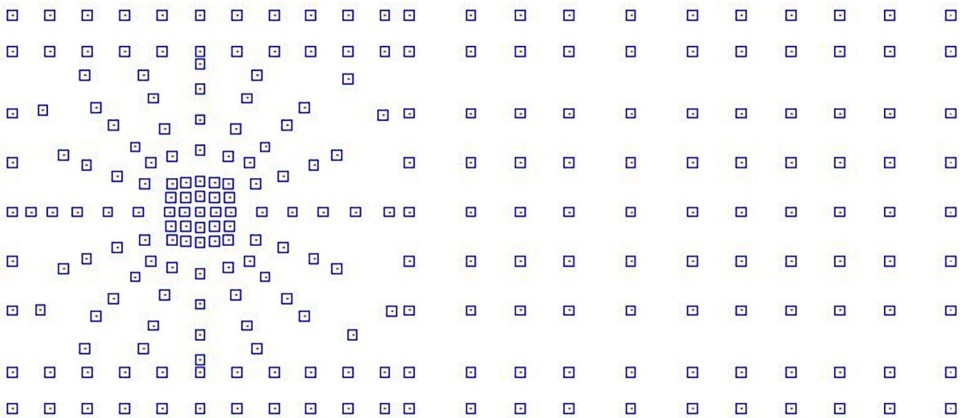

**Fig 7. Layout of isolation bearings.**

As for the vertical stiffness, it is often necessary to consider the load-bearing requirements and the precision of the natural period, typically taking at least 1000 times the horizontal stiffness [30,31]. Comprehensively considering the construction of the structure's baseplate, to avoid local stress concentration and excessive torsional vibration components, the structure is equipped with 247 isolation bearings at its base, arranged as shown in Fig 7 with parameters listed in the Table 3.

## Seismic input

As a novel approach to utilizing nuclear energy following nuclear power generation, there are currently no specific seismic design standards internationally for the swimming pool reactor. Therefore, The seismic waves used in this paper are the improved version of the Regulatory Guide 1.60 seismic design response spectrum developed by the US Nuclear Regulatory Commission in 1973, for earthquake response analysis based on the seismic safety review standards for nuclear power plants. The corresponding response spectrum is a broad-band spectrum that encompasses various damping conditions and is conservatively designed to typically envelop the response spectra of specific or expected sites in the country, providing a good characterization of the seismic capacity for nuclear facilities. The total duration is 28 seconds with a time step of 0.01 seconds. The PGA values are 0.3g in the X (Fig 8A) and Z directions (Fig 8C), and 0.2g in the Y direction (Fig 8B).

## Modal analysis comparison

In contrast to many existing studies focused on seismic mitigation and isolation of buildings, this paper considers SSI effects in the seismic resistance and isolation numerical simulation of the swimming pool reactor. Thus, utilizing computational modal analysis, this research distinctly delineates the impact of the isolation layer on the structure's overall vibration modes and actual responses. Furthermore, it examines the foundation's presence (SSI effect) on the

Table 3. Parameters of rubber isolation bearings.

| Vertical Force (kN) | Vertical Stiffness (N/m) | Horizontal Stiffness (N/m) | Horizontal Damping Coefficient (s/m) |
|---|---|---|---|
| 938 | $2.6 \times 10^9$ | $9.24896 \times 10^5$ | $1.76642 \times 10^5$ |

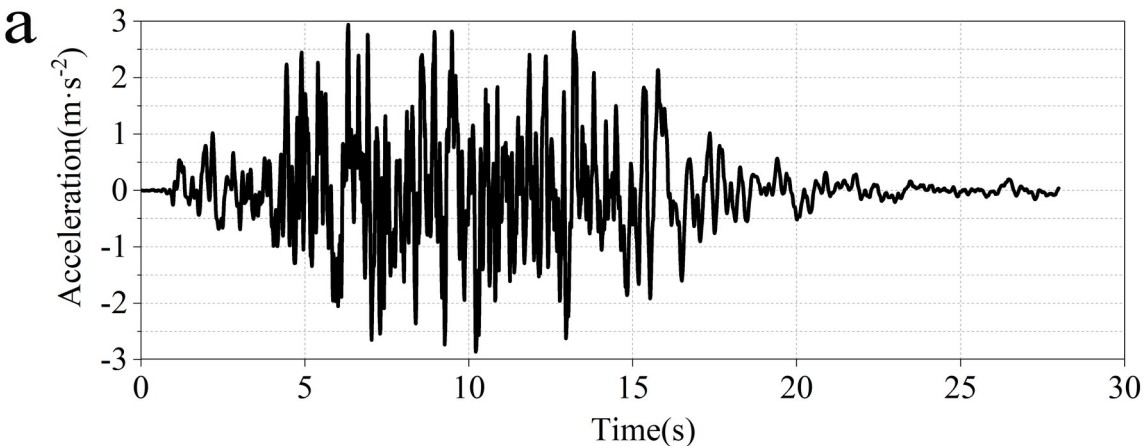

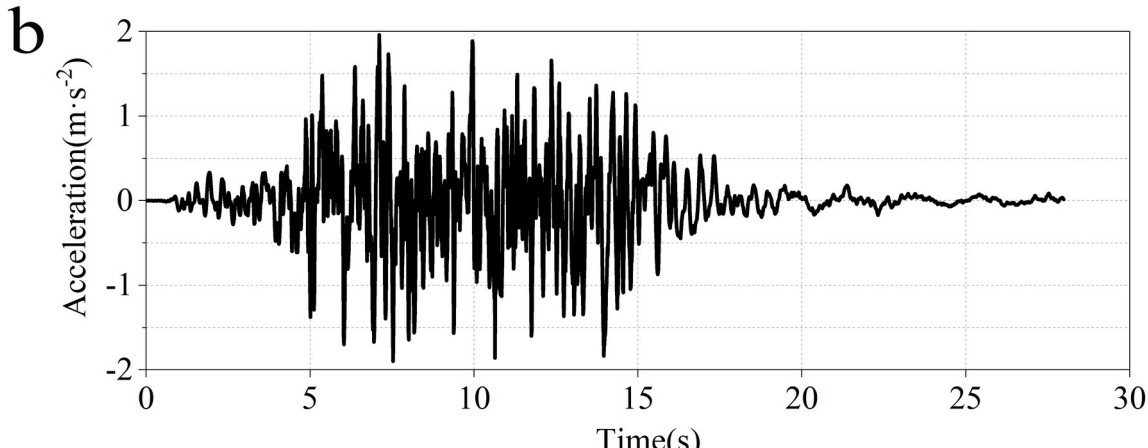

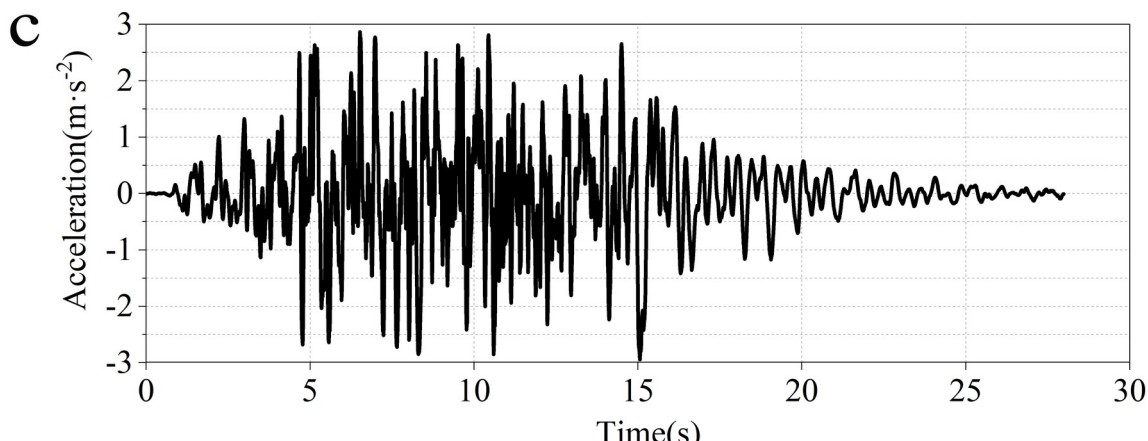

**Fig 8. Ground motion time-history curve of improved RG1.60. a**:X-direction. **b**: Y-direction. **c**: Z-direction.

structure's dynamic characteristics. The first-order natural vibration frequencies and periods before and after isolation under four different working conditions are listed in Table 4.

Analysis of the data in Table 4 reveals that:

**Table 4. Comparison of first-order natural vibration frequency and natural vibration period of the structure before and after isolation.**

| Condition | Natural Frequency (Hz) | | Isolated Model(s) | |
|---|---|---|---|---|
| | Non-Isolated Model | Isolated Model | Non-Isolated Model | Isolated Model |
| Condition 1 | 8.17080 | 0.50031 | 0.12239 | 1.99876 |
| Condition 2 | 7.47250 | 0.50017 | 0.13382 | 1.99932 |
| Condition 3 | 6.20450 | 0.49981 | 0.16117 | 2.00076 |
| Condition 4 | 1.53850 | 0.48836 | 0.64998 | 2.04767 |

1. Regardless of the foundation condition corresponding to each scenario, after adopting isolation measures for the swimming pool reactor, the structure's natural frequency significantly decreases, in other words, the natural period of the structure is substantially extended. Specifically, in Conditions 1, 2, 3, and 4, the natural frequencies are reduced by 93.9%, 93.3%, 91.9%, and 68.3%, respectively, with the natural periods increasing by approximately 16.33-fold, 15-fold, 12.4-fold, and 3.15-fold, respectively. The most effective isolation is observed under Condition 1, while it is comparatively less effective under Condition 4.

2. The comparison of first-order natural vibration frequency and natural vibration period of the structures after isolation with different conditions (i.e., 1 to 4) is not different. As a result, it is expected that the response of isolated structures under different conditions is similar. It is evident that under the influence of SSI effects, it becomes increasingly flexible with the gradual weakening of the rock properties and decreasing stiffness of the foundation. This flexibility shifts the vibration of the isolation structure system toward a lower frequency range, extending the natural period. This demonstrates that the hardness or softness of the foundation soil directly influences the natural period of the isolation structure system, thereby indirectly affecting its isolation effectiveness [32,33]. In summary, the feasibility of the simulation and application methods of the isolation layer proposed in this study is confirmed.

## Comparison of seismic response analysis

This study conducts a comparative study of the seismic responses of non-isolated and isolated structures of the swimming pool reactor under different foundation conditions. Typical indicators such as floor response spectrum, acceleration, sloshing wave height, and base shear are selected to assess the isolation layer's impact on the structure's seismic performance. Additionally, the seismic reduction rate $\Psi$ (SR) [34] is used as a quantitative evaluation index to measure the effectiveness of isolation. The formula is as follows:

$$\Psi = \frac{\delta - \Delta}{\delta} \times 100\% \tag{14}$$

In the formula, $\delta$ represents the seismic response before isolation, and $\Delta$ represents the seismic response after isolation. If $\Psi > 0$, it indicates that the isolation measures have reduced the seismic response; conversely, it can be inferred that the isolation measures further stimulate and amplify the seismic response.

### Acceleration analysis

To analyze the variation in structural acceleration along the vertical axis, nodes on the structure's sidewall, as depicted in Fig 9 are selected for result extraction. The maximum acceleration values at each node are obtained based on numerical analysis (Fig 10).

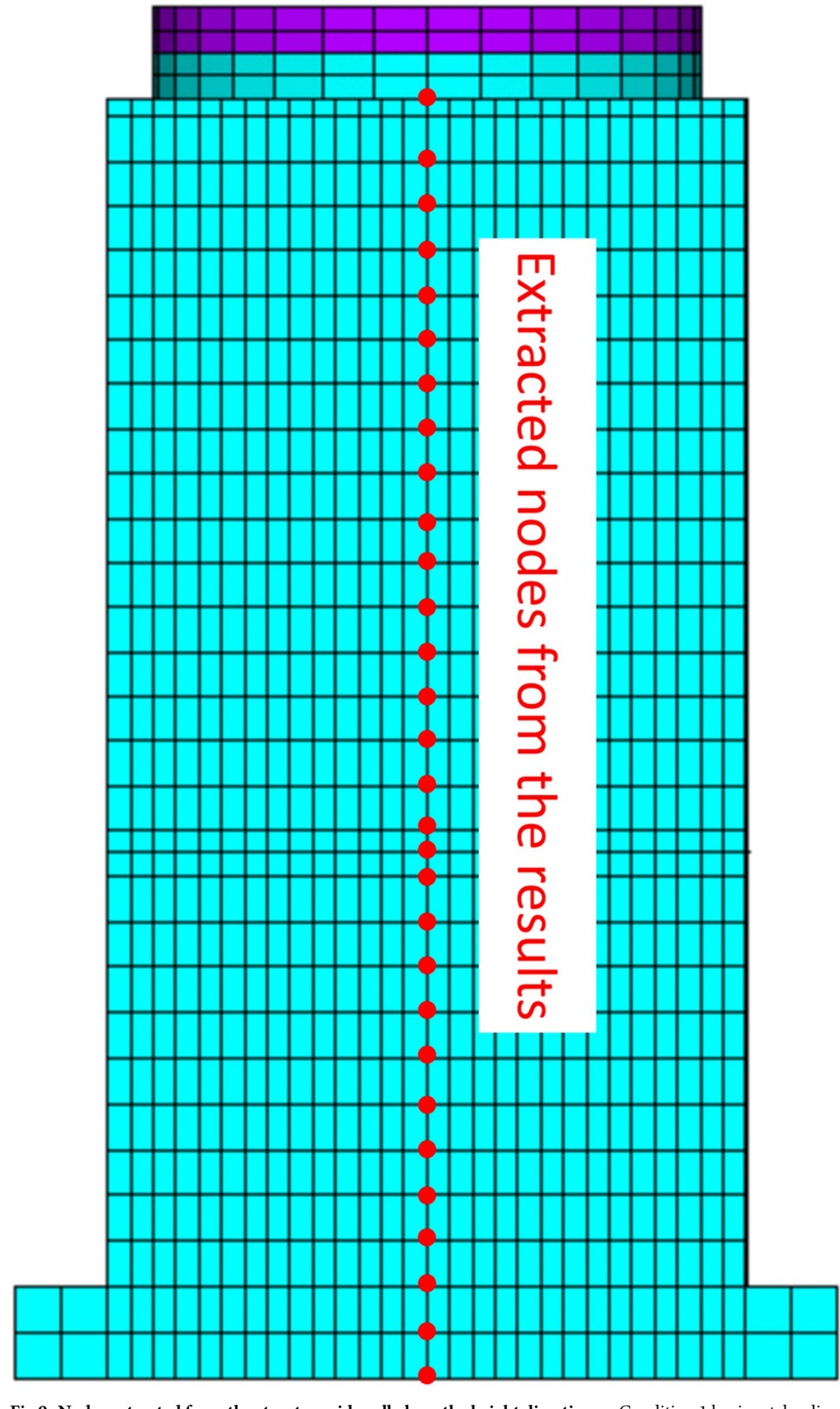

**Fig 9. Nodes extracted from the structure sidewall along the height direction. a**: Condition 1 horizontal x-direction. **b**: Condition 1 horizontal y-direction. **c**: Condition 2 horizontal x-direction. **d**: Condition 2 horizontal y-direction. **e**: Condition 3 horizontal x-direction. **f**: Condition 3 horizontal y-direction. **g**: Condition 4 horizontal x-direction. **h**: Condition 4 horizontal y-direction.

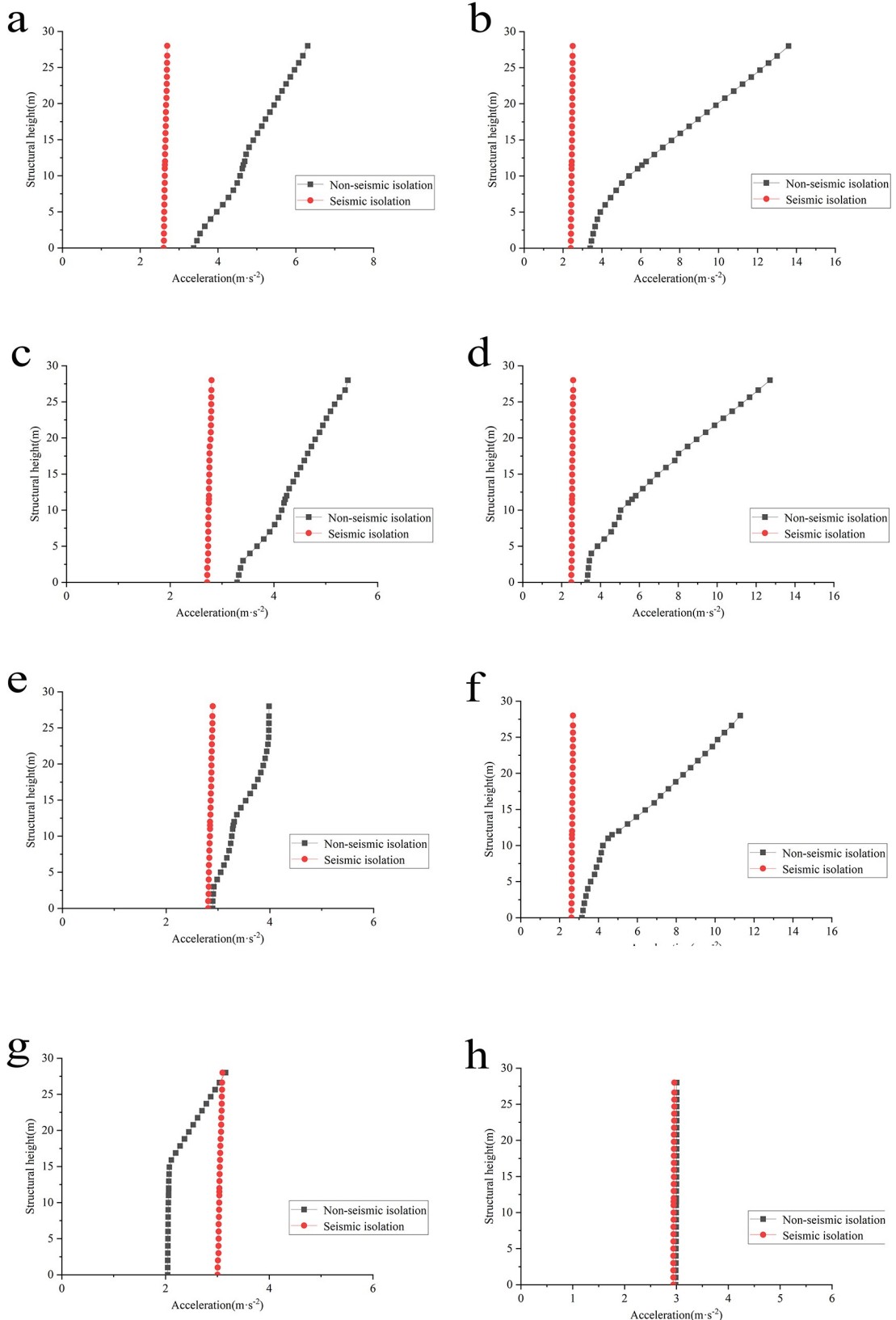

**Fig 10. Envelope curve of maximum acceleration distribution along the height of the sidewall under various working conditions. a**: Condition 1 horizontal x-direction. **b**:Condition 1 horizontal y-direction. **c**: Condition 2 horizontal x-direction. **d**: Condition 2 horizontal y-direction. **e**: Condition 3 horizontal x-direction. **f**: Condition 3 horizontal y-direction. **g**: Condition 4 horizontal x-direction. **h**: Condition 4 horizontal y-direction.

As can be seen from the following comparative analysis diagrams, the horizontal X-direction, before isolation, the peak accelerations of the swimming pool reactor occur at the top extraction point in all four different foundation conditions, with values of 6.3076m/s$^2$, 5.4270m/s$^2$, 3.9846m/s$^2$, and 3.1587m/s$^2$, respectively. After implementing base isolation, unlike the "diagonal increase" (i.e., The height of the structure and the acceleration of the structure increase at the same time) trend before isolation, the maximum acceleration values at various heights become more uniform, fluctuating only within a narrow range. The maximum and minimum acceleration differences in Conditions 1 to 4 do not exceed 0.1m/s$^2$. The structure shows an overall translational motion, with the highest acceleration values at the top being 2.6953m/s$^2$, 2.7995 m/s$^2$, 2.8981m/s$^2$, and 3.0983m/s$^2$, respectively, corresponding to reduction rates of approximately 57.3%, 48.4%, 27.3%, and 1.9% compared to the non-isolated response. The trend of acceleration changes in the horizontal Y-direction before and after isolation is similar to that in the X-direction, with reduction rates at the top extraction point in Conditions 1 to 4 of 81.6%, 79.6%, 76.2%, and 1.5%, respectively.

The above analysis indicates that the steel cover plate at the top of the reactor pool, positioned at the highest elevation, exhibits the highest acceleration response, marking it as the most vulnerable component of the structure with the most significant risk of damage. Consequently, acceleration time-history curves are extracted and compared before and after isolation under the four foundation conditions (Fig 11).

The acceleration time-course response illustrated in Fig 11 demonstrates that structural acceleration decreases to varying degrees across all four conditions following the application of the isolation layer in both horizontal X and Y directions. This reduction is most pronounced under the first condition (Fig 11A and 11B), indicating optimal damping effectiveness. However, as the analysis progresses through the remaining three conditions, the reduction in acceleration and the corresponding damping effect diminish, consistent with the previous paragraph's analysis of peak accelerations. From this analysis, it can be concluded that the SSI effect on the seismic isolation performance of the swimming pool reactor is intricately linked to the characteristics of the foundation soil. As the soil stiffness significantly decreases when transitioning from lithological to softer soil foundations, the damping effect on the isolated structure's acceleration diminishes. In simpler terms, the damping rate is higher on hard foundations than on soft ones.

## Analysis of floor response spectrum

The implementation of base isolation has extended the natural vibration period of the swimming pool reactor, significant changes in the floor response spectrum. Thus, this study continues to use the steel cover plate as the research object, extracting the acceleration response spectrum at this location for investigation and analysis. It presents the floor response spectrum of the steel cover plate before and after isolation under the four foundation conditions (Fig 12).

Taking Condition 1 (Fig 12A and 12B) as an example, in both horizontal directions, within the low-frequency range (0.2–0.6 Hz), the floor response spectra of the isolated structure are generally higher than those of the non-isolated model. This is precisely because the base isolation layer extends the structure's natural period to reduce seismic effects, which may not be effective for long-period (low-frequency) structures. At the same time, the maximum response

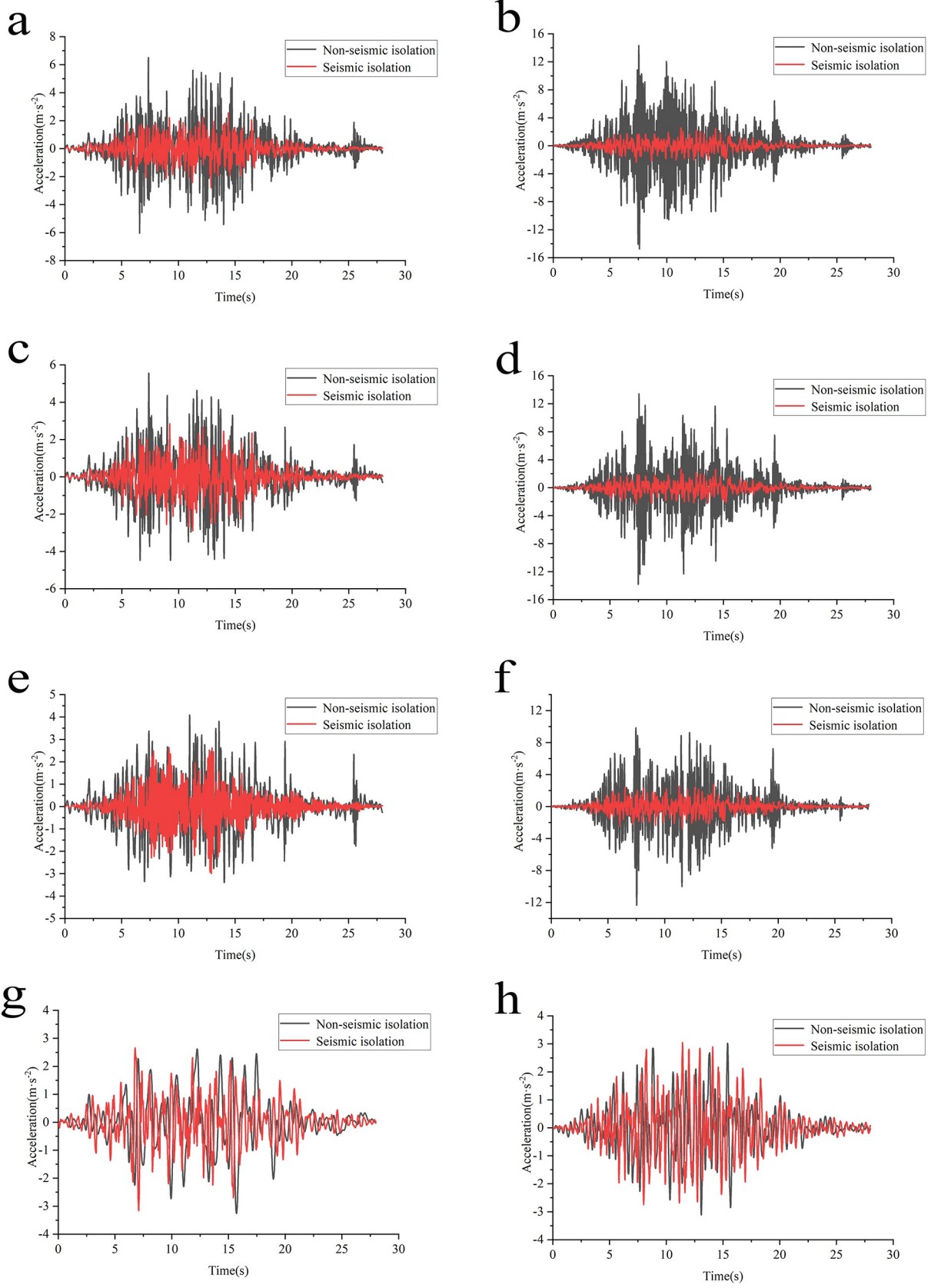

**Fig 11. Acceleration time-history curve of steel cover plate under various working conditions. a**: Condition 1 horizontal x-direction. **b**: Condition 1 horizontal y-direction. **c**: Condition 2 horizontal x-direction. **d**: Condition 2 horizontal y-direction. **e**: Condition 3 horizontal x-direction. **f**: Condition 3 horizontal y-direction. **g**: Condition 4 horizontal x-direction. **h**: Condition 4 horizontal y-direction.

spectra values occur around 0.5 Hz, corresponding to the isolation period of the structure. In the frequency range above 0.6 Hz (high-frequency range), the floor response spectrum of the isolated structure is significantly lower than that of the non-isolated structure, indicating that the isolation system effectively mitigates seismic effects. The swimming pool reactor shows similar patterns under the other three conditions, with slight numerical differences.

In the horizontal X-direction, the peak values of the acceleration response spectra in Conditions 1 to 4 for the isolated structure are $12.30\text{m/s}^2$, $9.34\text{m/s}^2$, $6.21\text{m/s}^2$, and $6.81\text{m/s}^2$, respectively, with reduction rates compared to the non-isolated structure of 57.02%, 48.72%, 40.29%, and 21.86%. In the Y-direction, the peak values are $14.21\text{m/s}^2$, $13.87\text{ m/s}^2$, $9.90\text{m/s}^2$, and $8.11\text{m/s}^2$, with reduction rates of 83.71%, 81.89%, 79.46%, and 12.09%, respectively. It is evident that as soil stiffness decreases, particularly when the lithological foundation gradually changes to the softer soil foundation, the damping effect of the isolated structure's floor response spectrum also gradually decreases.

## Displacement analysis

Based on the results of extracting points along the height direction of the side wall of the swimming pool reactor mentioned above, list the distribution curve of the maximum horizontal displacement response of the swimming pool reactor relative to the base before and after isolation along the height of the side wall (Fig 13).

The figure illustrates that the structure's maximum relative displacement occurs at the top of the side wall, regardless of whether it is isolated. Compared to non-isolated cases, the displacement of the structure relative to the base exhibits varying degrees of reduction across all four foundation conditions in both horizontal X and Y directions after seismic isolation. Specifically, before isolation, the maximum relative displacements in the horizontal X direction for Conditions 1 through 4 are 0.24mm, 0.3mm, 0.36mm, and 40.18mm, respectively. After isolation, these displacements are reduced to 0.15mm, 0.21mm, 0.26mm, and 30.99mm, corresponding to reduction rates of 37.5%, 30%, 27.8%, and 22.9%, respectively. In the horizontal Y direction, the maximum relative displacements before isolation are 1.45mm, 1.63mm, 3.6mm, and 5.8mm. After isolation, they are 0.84mm, 1.01mm, 2.41mm, and 4.15mm, with reduction rates of 41.8%, 37.7%, 33.1%, and 28.5%, respectively. This reduction is attributed to the installation of the isolation layer, which imparts lower stiffness characteristics between the structural base and the raft foundation. This characteristic enables significant translational deformation of the isolation layer, dissipating upward seismic energy and reducing structural deformation in both directions, thereby enhancing the isolation effect. Additionally, the relative displacement progressively decreases from Condition 1 to 4, indicating that the isolation effect is closely related to the nature of the foundation soil. The data suggest isolation is more effective on lithological foundations than medium-soft soil foundations.

## Base shear analysis

Base shear is emphasized in numerous technical guidelines and standards as an indispensable indicator of seismic response. This section introduces base shear as a measure to assess isolation effectiveness, presenting time-history curves of base shear before and after isolation under four different working conditions (Fig 14).

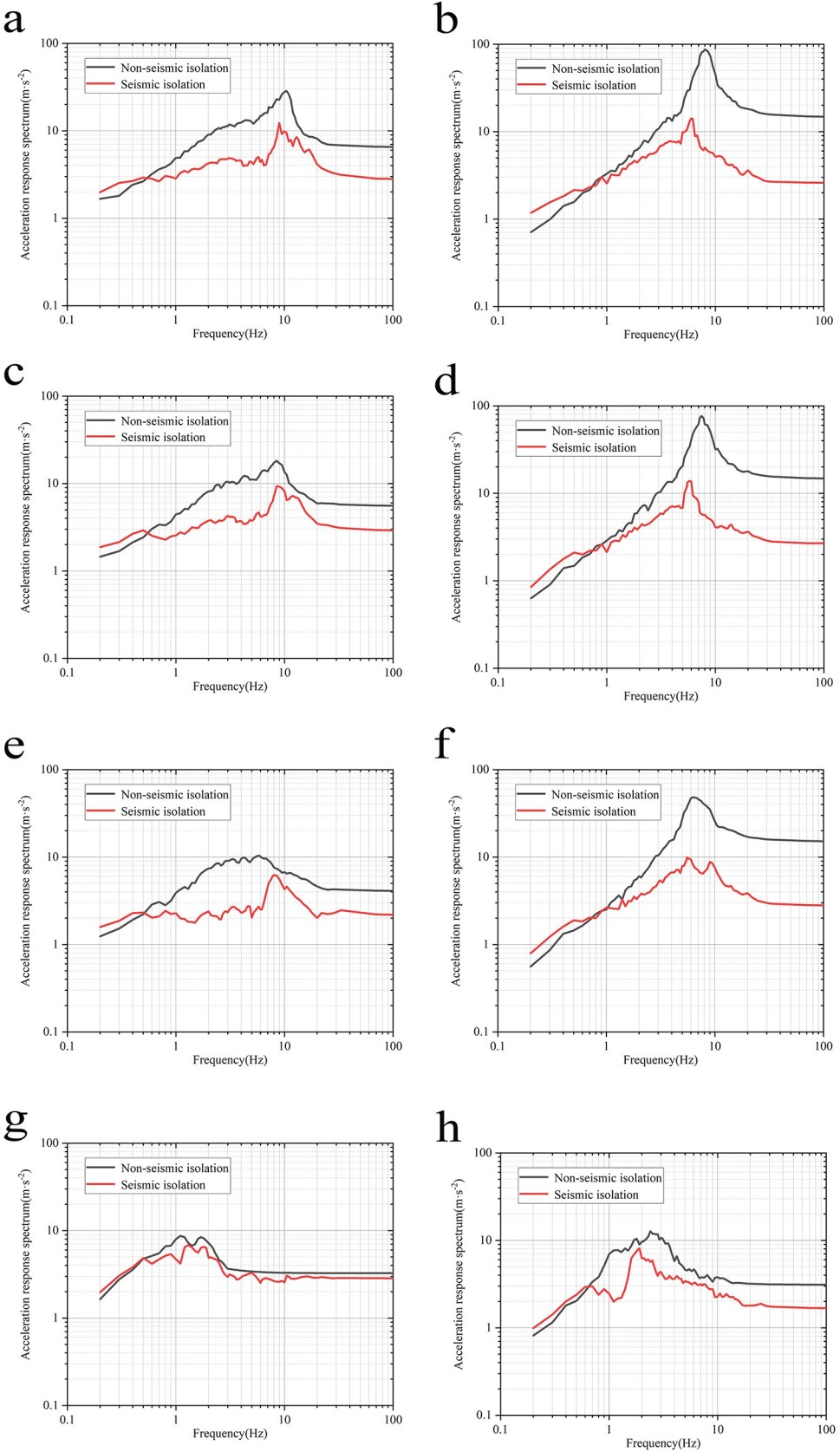

**Fig 12. Floor response spectrum of steel cover plate under various working conditions. a**: Condition 1 horizontal x-direction. **b**:Condition 1 horizontal y-direction. **c**: Condition 2 horizontal x-direction. **d**: Condition 2 horizontal y-direction. **e**: Condition 3 horizontal x-direction. **f**: Condition 3 horizontal y-direction. **g**: Condition 4 horizontal x-direction. **h**: Condition 4 horizontal y-direction.

Fig 14 reveals that the swimming pool reactor demonstrates a similar pattern of change under all four foundation conditions. In the non-isolated structure, the base shear fluctuates dramatically and varies steeply over time. In contrast, the isolated structure's base shear has a relatively low variation frequency, with a smoother curve and overall values consistently lower than those of the non-isolated structure. This is due to the isolation layer effectively lowering the structure's natural frequency and imparting appropriate damping characteristics, significantly reducing the acceleration generated by the structure itself, thus effectively weakening the seismic forces. The peak base shear response reduction varies with different foundation conditions, with the most significant reduction in Condition 1 at 70.79% and decreasing sequentially in the subsequent conditions to 69.83%, 59.14%, and 27.38%. Similarly, the horizontal Y-direction demonstrates the same pattern. From conditions 1 to 4, as the foundation rigidity decreases, the isolation rate also gradually reduces, meaning that the isolation effect on lithological foundations is superior to that on medium-soft-soil foundations.

## Sloshing wave height analysis

Sloshing wave heights in the reactor pool and spent fuel pool, before and after isolation under each working condition (Fig 15).

1. After isolation, the time-history curves of wave heights in the reactor pool under all four working conditions align in trend and linearity, with only minor numerical differences, and a similar pattern is observed for the spent fuel pool. Notably, the wave height time-history curve before isolation in Condition 4 exhibits more severe fluctuations than the other three conditions. After isolation, the violent fluctuations are avoided, lowering the frequency of curve changes and smoothing the overall curve. This demonstrates that implementing the base isolation layer maintains consistent vibration behavior of the swimming pool reactor under different foundation conditions, thus minimizing the impact of foundation conditions (medium-soft-soil) on the liquid sloshing behavior after isolation. The peak values of wave height responses are higher after isolation in all conditions, showing that isolation measures further stimulate and amplify the wave height responses, with a prominent delay in the timing of peak values. However, the wave height responses are still within the safe margins set for the reactor and spent fuel pools, ensuring no overflow or leakage occurs and maintaining the structural integrity and reliability of the swimming pool reactor.

2. For the non-isolated structure, the peak wave heights in the reactor pool across Conditions 1 to 4 are 0.82324m, 0.82551m, 0.84159m, and 0.98561m, respectively. After isolation, these values increased to 1.12940m, 1.13040m, 1.13130m, and 1.19450m, resulting in negative reduction rates of -37.19%, -36.93%, -34.42%, and -21.19%. This indicates that, regardless of the foundation condition, the sloshing wave height responses are significantly greater after isolation, suggesting that the isolation measures further amplified the wave height responses. Moreover, the timing of peak wave heights in the four working conditions shifts from 12.32s, 12.33s, 12.34s, and 12.51s before isolation to 13.96s, 13.96s, 13.98s, and 13.93s after isolation, highlighting a significant hysteresis effect. The spent fuel pool exhibited a similar pattern in peak wave heights and timing post-isolation, with only numerical differences. This analysis suggests that the implementation of the isolation layer has an

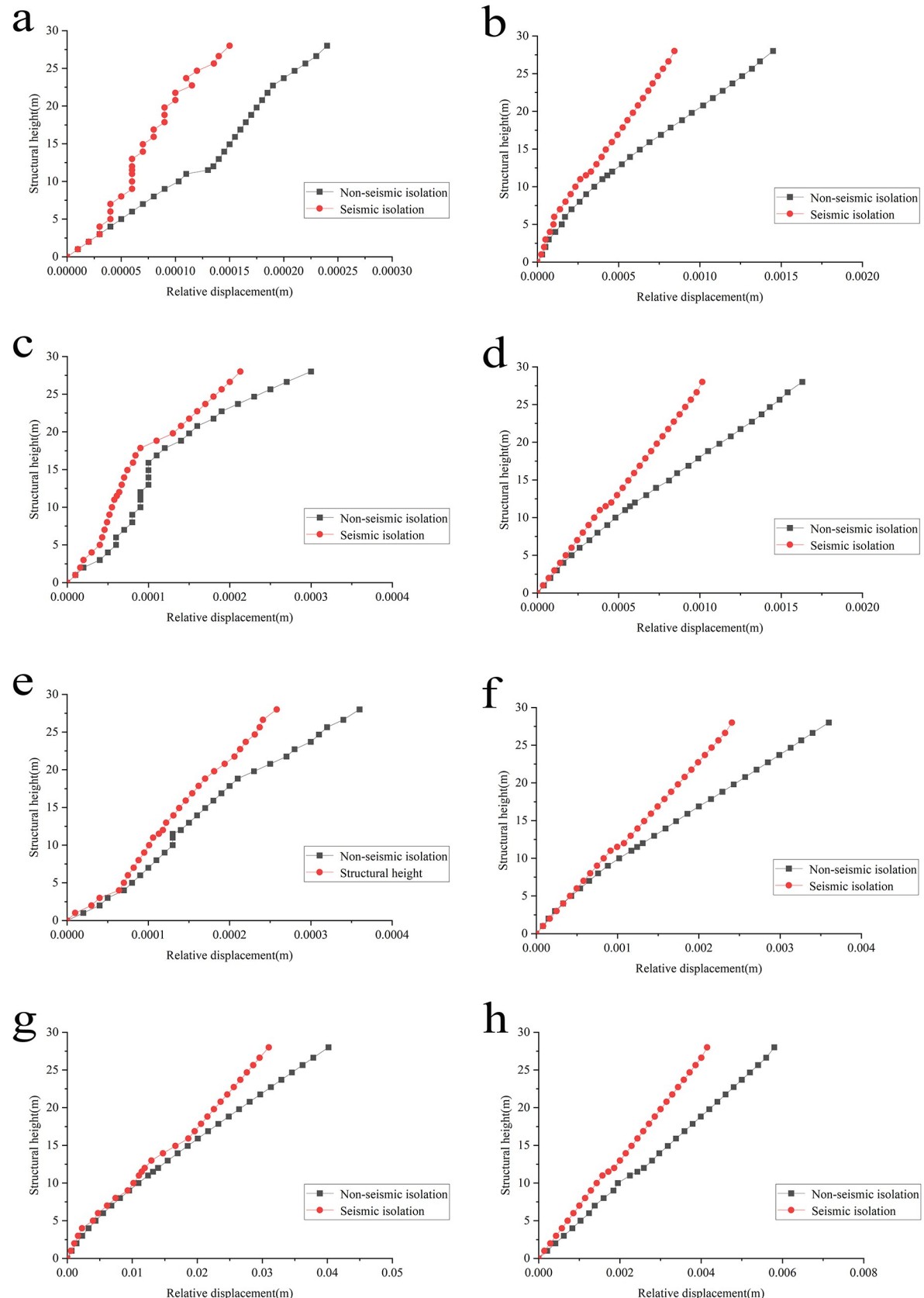

**Fig 13. Distribution curves of maximum relative displacement along sidewall height under various working conditions. a**: Condition 1 horizontal x-direction. **b**: Condition 1 horizontal y-direction. **c**: Condition 2 horizontal x-direction. **d**: Condition 2 horizontal y-direction. **e**: Condition 3 horizontal x-direction. **f**: Condition 3 horizontal y-direction. **g**: Condition 4 horizontal x-direction. **h**: Condition 4 horizontal y-direction.

amplifying effect on sloshing wave heights due to the low-frequency nature of liquid sloshing, which coincides with the long-period characteristics of the isolation layer, further stimulating the liquid's dynamic behavior.

3. Seismic isolation led to varying increases in sloshing wave heights. After isolation, the highest wave heights in the reactor pool for Conditions 1 to 4 occurred at heights of 1.8706m, 1.8696m, 1.8687m, and 1.8055m from the top of the container, respectively. The highest wave heights in the spent fuel pool occurred at 1.0589m, 1.0575m, 1.0559m, and 0.9937m, respectively. The wave height responses remained within the control limits and did not exceed the sloshing safety margins established for both the surge and spent fuel pools. Therefore, there is no liquid overflow or leakage risk, ensuring that the swimming pool reactor structure remains safe and reliable.

## Conclusion

To investigate the seismic performance of base-isolated swimming pool reactor structures under varying foundation conditions, this paper considers the dynamic hydraulic effects within storage containers and the impacts of Soil-Structure Interaction (SSI). A structure-foundation calculation model for the swimming pool reactor is developed using the viscous-spring artificial boundary. Local base isolation technology is used to examine the seismic isolation effects on the structure and the influence of various foundation conditions on the dynamic response of the base-isolated structure. Based on the analysis and discussion of typical indicator data, the following conclusions are drawn:

1. Regardless of foundation conditions, adopting base isolation measures for the swimming pool reactor significantly reduces its natural frequency, and the magnification factor of the extended natural period decreases as the foundation transitions from stiff to soft. Under SSI effects, as conditions shift from Condition 1 to Condition 4, the foundation becomes progressively softer, with its rockiness weakening and stiffness decreasing. This leads to the base-isolated structure's vibrations progressively moving toward lower frequency ranges. The base-isolated structure's natural frequency decreases incrementally, indicating that the foundation soil's stiffness level indirectly affects the isolation effect by altering the natural period of the base-isolated structure system.

2. Analyses of acceleration, floor response spectra, displacement, and base shear force revealed that under SSI effects, isolation effectiveness also decreased as foundation stiffness declined across the four woeking conditions, with superior performance on lithological foundations compared to medium-soft soil foundations.

3. After implementing base isolation measures, sloshing wave height responses significantly increase across all foundation conditions. This observation indicates that isolation measures further stimulate and amplify the wave height response. Moreover, a distinct hysteresis effect is evident in the timing of peak wave heights. However, the wave height responses generated across all four working conditions remain manageable and have stayed within the sloshing safety margins allocated for the reactor and spent fuel pools. Therefore, liquid

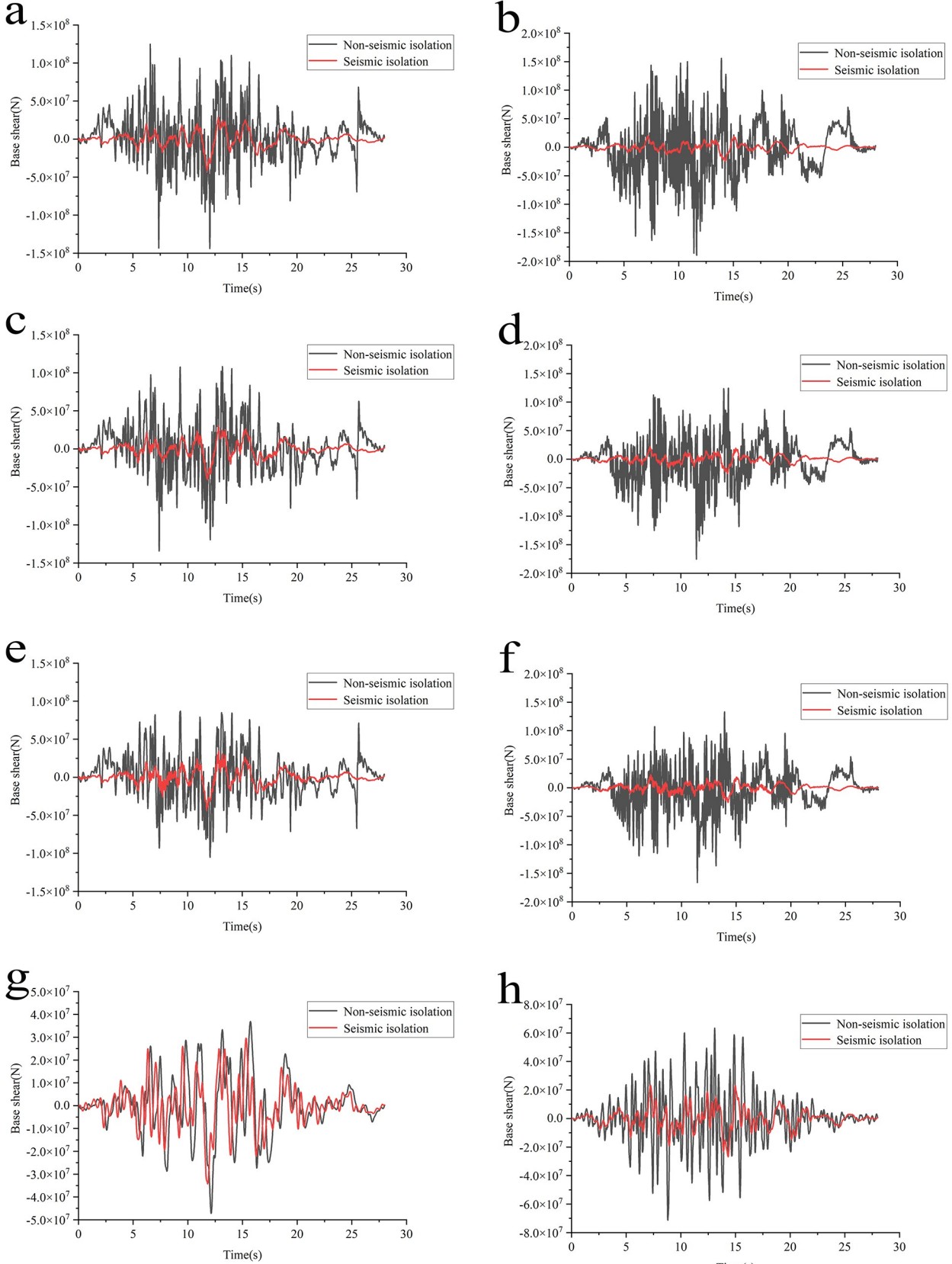

**Fig 14. Comparison of base shear time-history before and after structural isolation under various working conditions. a**: Condition 1 horizontal x-direction. **b**: Condition 1 horizontal y-direction. **c**: Condition 2 horizontal x-direction. **d**: Condition 2 horizontal y-direction. **e**: Condition 3 horizontal x-direction. **f**: Condition 3 horizontal y-direction. **g**: Condition 4 horizontal x-direction. **h**: Condition 4 horizontal y-direction.

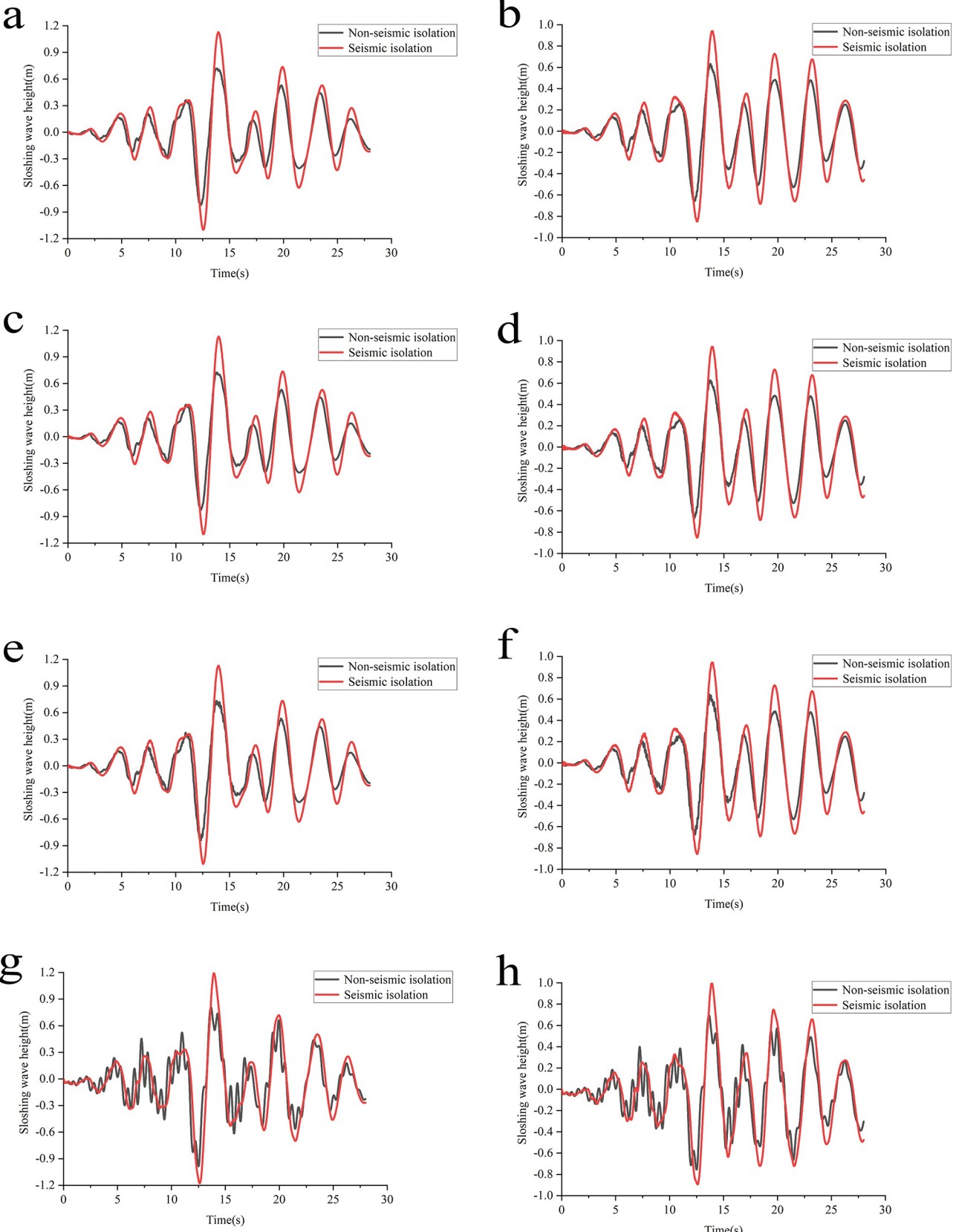

**Fig 15. Time-history curves of sloshing wave height under various working conditions. a**: Condition 1 reactor pool. **b**: Condition 1 spent fuel pool. **c**: Condition 2 reactor pool. **d**: Condition 2 spent fuel pool. **e**: Condition 3 reactor pool. **f**: Condition 3 spent fuel pool. **g**: Condition 4 reactor pool. **h**: Condition 4 spent fuel pool.

spills or leaks are negligible, ensuring the swimming pool reactor structure remains safe and reliable.

## Supporting information

**S1 File. Applying viscoelastic boundary and seismic motion input calculation program code.**
(ZIP)

## Acknowledgments

The authors extend their thanks to Qian Bo and Yanli Peng for technical assistance with the ANSYS.

## Author Contributions

**Conceptualization:** Jie Zhao, Jianshan Wang.

**Data curation:** Jiehua Huang.

**Formal analysis:** Jianshan Wang.

**Funding acquisition:** Jie Zhao.

**Investigation:** Jiehua Huang.

**Methodology:** Jie Zhao.

**Project administration:** Jie Zhao.

**Resources:** Jie Zhao.

**Software:** Jie Zhao, Jiehua Huang.

**Supervision:** Jie Zhao.

**Validation:** Jianshan Wang.

**Visualization:** Jianshan Wang.

**Writing – original draft:** Jianshan Wang, Jiehua Huang.

**Writing – review & editing:** Jie Zhao, Jianshan Wang.

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
