## [Decision Letter · Decision Letter 0]

22 Jul 2024

PONE-D-24-23674Seismic performance of base-isolated structures for swimming pool reactors under Different foundation conditionsPLOS ONE

Dear Dr. Zhao,

Thank you for submitting your manuscript to PLOS ONE. After careful consideration, we feel that it has merit but does not fully meet PLOS ONE’s publication criteria as it currently stands. Therefore, we invite you to submit a revised version of the manuscript that addresses the points raised during the review process.

**ACADEMIC EDITOR: **

We look forward to receiving your revised manuscript.

Kind regards,

Mohammadreza Vafaei, Ph.D.

Academic Editor

PLOS ONE

Journal Requirements:

Reviewers' comments:

Reviewer's Responses to Questions

**Comments to the Author**

1. Is the manuscript technically sound, and do the data support the conclusions?

Reviewer #1: Yes

Reviewer #2: Yes

2. Has the statistical analysis been performed appropriately and rigorously? 

Reviewer #1: Yes

Reviewer #2: Yes

3. Have the authors made all data underlying the findings in their manuscript fully available?

Reviewer #1: Yes

Reviewer #2: Yes

4. Is the manuscript presented in an intelligible fashion and written in standard English?

Reviewer #1: Yes

Reviewer #2: Yes

5. Review Comments to the Author

Reviewer #1: Comments to the Author

I have reviewed the manuscript “Seismic performance of base-isolated structures for swimming pool reactors under Different foundation conditions” by Jie Zhao,Jianshan Wang,Jiehua Huang. This is an interesting study and needs some revisions to be published in Asian Journal of Civil Engineering. Here are my comments.

1. Please, revise some typos in the text and figures (e.g., Figure 1).

2. Within Introduction and text, discuss that a numerical model is always affected by two families of uncertainties (i.e., aleatory and epistemic), as discussed in:

Gino, D. et al., Strain-based method for assessment of global resistance safety factors for NLNAs of reinforced concrete structures, Engineering Structures, 2024, 304, 117625.

Miceli, E. et al., Approaches to estimate global safety factors for reliability assessment of RC structures using non-linear numerical analyses, Engineering Structures, 2024, 311, 118193.

3. Please, provide numerical details (e.g., criteria convergence).

4. Comments if the concrete confinement effects are considered, as discussed in:

Miceli, E., et al., Confinement effects within the seismic design of reinforced concrete frames: reliability assessment and comparison, Engineering Structures, 2024, 313, 118248.

5. Please, provide the inherent damping factors used in the numerical analyses.

6. Discuss if the records can be considered frequent or no-frequent ground motions as well as their PGA/PGV ratios, as described in:

Castaldo, P., Miceli, E. Optimal single concave sliding device properties for isolated multi-span continuous deck bridges depending on the ground motion characteristics, Soil Dynamics and Earthquake Engineering, 2023, 173, 108128.

Reviewer #2: The research topic was a swimming pool reactor (SPR) building, and ANSYS software was used to create a 3D dynamic interaction model that included the liquid sloshing effect and the secondary development characteristics of user-programmable features (UPFs). Viscos-spring boundary elements were used to account for the energy dissipation from scattered waves, and the Housner equivalent mechanical model was used to model the dynamic hydraulic effect. This study investigates the influence of isolation measures on the seismic mitigation performance of the structure, taking into account the effects of soil-structure interaction (SSI). It looks into how different foundation circumstances impact the isolated structure's ability to withstand earthquakes. The findings show that as site stiffness decreases, the seismic isolation ratios for base shear, acceleration, floor response spectra, and displacement decrease. The paper is interesting, and the following are some specific observations:

1. The comparison of first-order natural vibration frequency and natural vibration period of the structures after isolation with different conditions (i.e., 1 to 4) is not different. As a result, it is expected the similar response of isolated structure with different conditions.

2. It is not clear what the authors are conveying through “diagonal increase” on line 311 of the manuscript.

3. Some more details about the elements used in the ANSYS model for the analysis shall be provided.

4. Authors must explain about the User Programmable Features (UPFs) and FORTRAN, subprograms written in FORTRAN interface with ANSYS to customize user elements (refer to lines 133-134).

5. The authors shall provide More details about the selected earthquake ground motions such as year, components, PGA, etc.

6. Some results shall be presented considering the real and recorded near-fault earthquake ground motions.

7. The model of the base-isolated structure (refer to Figure 1) in the present study considers the flexibility of the superstructures and ignores the impact at the base on the adjacent moat wall. The above assumption for the selected model shall be justified by citing suitable papers like https://doi.org/10.1155/2003/368693 , https://doi.org/10.1177/1077546309103271 , and others.

8. The presentation of the paper needs to be further improved to meet international journal standards.

6. PLOS authors have the option to publish the peer review history of their article (what does this mean?). If published, this will include your full peer review and any attached files.

Reviewer #1: **Yes: **Paolo Castaldo

Reviewer #2: No

---

## [Author Response · Author response to Decision Letter 0]

24 Aug 2024

Title: Seismic performance of base-isolated structures for swimming pool reactors under Different foundation conditions

Authors: Jie Zhao, Jianshan Wang, Jiehua Huang

Paper No.: PONE-D-24-23674

Dear Editor,

The authors would like to thank the reviewers for their valuable comments and suggestions. The reviewers’ comments have all been taken into consideration in this revised manuscript. The authors believe that these suggestions significantly improve the quality of this paper. The following presents our replies to the questions and concerns that are raised by the Reviewers. 

Reply to Reviewer 1’s Comments:

1. Please, revise some typos in the text and figures (e.g., Figure 1).

Reply: Agreed, thank you for the expert's comments. The revisions have been made in the manuscript.

2. Within Introduction and text, discuss that a numerical model is always affected by two families of uncertainties (i.e., aleatory and epistemic), as discussed in:

Gino, D. et al., Strain-based method for assessment of global resistance safety factors for NLNAs of reinforced concrete structures, Engineering Structures, 2024, 304, 117625.

Miceli, E. et al., Approaches to estimate global safety factors for reliability assessment of RC structures using non-linear numerical analyses, Engineering Structures, 2024, 311, 118193.

Reply: Agreed, thank you for the expert's comments. The main research focus of this paper is the seismic performance study of base isolation structures for pool-type reactors, and the developed viscoelastic boundary has been validated through case studies, proving its accuracy. The expert's suggestions are very helpful, and I will consider the two types of uncertainties in numerical models (i.e., aleatory and epistemic) more in future research.

3. Please, provide numerical details (e.g., criteria convergence).

Reply: Agreed, thank you for the expert's comments. The force convergence criterion for dynamic calculations is 0.05, and the displacement convergence criterion is also 0.05. The soil is modeled using the Mohr-Coulomb model, the concrete structure is considered linear elastic. The paper employs a direct coupling method for fluid and solid, enforcing the equality of degrees of freedom at coincident nodes on the fluid-solid interface to achieve displacement transfer.

4. Comments if the concrete confinement effects are considered, as discussed in:

Miceli, E., et al., Confinement effects within the seismic design of reinforced concrete frames: reliability assessment and comparison, Engineering Structures, 2024, 313, 118248.

Reply: Agreed. Thank you for the expert's comments. The paper does not consider the specific constraint effects of concrete structures, which is a very good suggestion. I will consider the concrete confinement effects of concrete structures in future research.

5. Please, provide the inherent damping factors used in the numerical analyses.

Reply: Agreed. Thank you for the expert's comments. According to Standard for Seismic Design of Nuclear Power Plants, the recommended inherent damping factor is 5%.

6. Discuss if the records can be considered frequent or no-frequent ground motions as well as their PGA/PGV ratios, as described in:

Castaldo, P., Miceli, E. Optimal single concave sliding device properties for isolated multi-span continuous deck bridges depending on the ground motion characteristics, Soil Dynamics and Earthquake Engineering, 2023, 173, 108128.

Reply: Agreed. Thank you for the expert's comments. Currently, there are no specific seismic design standards internationally. Therefore, the Standards for the Evaluation of Seismic Safety of Nuclear Power Plants are followed. The seismic waves used in this paper are the improved version of the Regulatory Guide 1.60 seismic design response spectrum developed by the US Nuclear Regulatory Commission in 1973, with a PGA value of 0.3g.

Reply to Reviewer 2’s Comments:

1. The comparison of first-order natural vibration frequency and natural vibration period of the structures after isolation with different conditions (i.e., 1 to 4) is not different. As a result, it is expected the similar response of isolated structure with different conditions. 

Reply: Agreed. Thank you for the expert's comments. The modification has been made in line 298 of the manuscript.

2. It is not clear what the authors are conveying through “diagonal increase” on line 311 of the manuscript.

Reply: Agreed. Thank you for the expert's comments. The “diagonal increase” in the manuscript means that both the structural height and the structural acceleration are increasing simultaneously.

3. Some more details about the elements used in the ANSYS model for the analysis shall be provided.

Reply: Agreed. Thank you for the expert's comments. In the numerical model analysis, SOLID185 elements were used to simulate the structure and foundation. Based on single-node definition, the software's built-in structural mass element MASS21 was chosen for simulation. For the effects of liquid in the water storage tank and spent fuel pool, FLUID80 elements were used based on the Housner theory, with coupled degrees of freedom used to simulate fluid-structure interaction effects. The foundation was divided into uniform grids as much as possible, with structural element sizes ranging from 0.5 to 2 meters, foundation element sizes shall not exceeding 5 meters. The model consists of 88,031 solid elements, 7,488 fluid elements, and 39 concentrated mass elements, totaling 95,558 elements and 111,171 nodes. 

4. Authors must explain about the User Programmable Features (UPFs) and FORTRAN, subprograms written in FORTRAN interface with ANSYS to customize user elements (refer to lines 133-134).

Reply: Agree, thanks to the expert's comments. the User Programmable Features (UPFs) serve as a supplement and extension to the APDL’s functionality. Essentially, it is a form of source code-level secondary development. Through a series of custom files (user subroutines) provided by ANSYS, users can edit and modify them, then use specific versions of compilers (MS C++ and FORTRAN) to generate a customized version of ANSYS software. This allows users to achieve desired functions or capabilities not provided by the software, such as optimization algorithms, user-defined failure criteria, and contact criteria, as well as the development of complex material constitutive models. 

5. The authors shall provide More details about the selected earthquake ground motions such as year, components, PGA, etc.

Reply: Agreed, and thank you for the expert's comments. Currently, there are no specific seismic design standards for this internationally. Therefore, The seismic waves used in this paper are the improved version of the Regulatory Guide 1.60 seismic design response spectrum developed by the US Nuclear Regulatory Commission in 1973, for earthquake response analysis based on the seismic safety review standards for nuclear power plants. The corresponding response spectrum is a broad-band spectrum that encompasses various damping conditions and is conservatively designed to typically envelop the response spectra of specific or expected sites in the country, providing a good characterization of the seismic capacity for nuclear facilities. The total duration is 28 seconds with a time step of 0.01 seconds. The PGA values are 0.3g in the X and Z directions, and 0.2g in the Y direction. 

6. Some results shall be presented considering the real and recorded near-fault earthquake ground motions.

Reply: Agree, thank you for the expert's comments. This paper uses the seismic design response spectrum for nuclear power plants. The expert's advice is excellent, and I will consider real and recorded near-fault earthquake motions in future research and present some results.

7. The model of the base-isolated structure (refer to Figure 1) in the present study considers the flexibility of the superstructures and ignores the impact at the base on the adjacent moat wall. The above assumption for the selected model shall be justified by citing suitable papers like https://doi.org/10.1155/2003/368693 , https://doi.org/10.1177/1077546309103271 , and others.

Reply: Thank you for the expert's comments. Figure 1 is a schematic representation of the static and dynamic models of viscoelastic artificial boundaries. According to the comments of experts, the base isolation structure has been improved in line 243.

8. The presentation of the paper needs to be further improved to meet international journal standards.

Reply: Agree, thank you for the expert's comments. The entire manuscript has been polished and improved.

Please feel free contact us if further revision is needed.

Thank you very much for your attention.

Best regards,

Dr. Zhao

School of Architectural Engineering

Dalian University

Dalian 116622

P. R. China

E-mail: zhaojie_gd@163.com

---

## [Decision Letter · Decision Letter 1]

3 Sep 2024

PONE-D-24-23674R1Seismic performance of base-isolated structures for swimming pool reactors under Different foundation conditionsPLOS ONE

Dear Dr. Zhao,

Thank you for submitting your manuscript to PLOS ONE. After careful consideration, we feel that it has merit but does not fully meet PLOS ONE’s publication criteria as it currently stands. Therefore, we invite you to submit a revised version of the manuscript that addresses the points raised during the review process.

**ACADEMIC EDITOR: **

We look forward to receiving your revised manuscript.

Kind regards,

Mohammadreza Vafaei, Ph.D.

Academic Editor

PLOS ONE

Journal Requirements:

Reviewers' comments:

Reviewer's Responses to Questions

**Comments to the Author**

1. If the authors have adequately addressed your comments raised in a previous round of review and you feel that this manuscript is now acceptable for publication, you may indicate that here to bypass the “Comments to the Author” section, enter your conflict of interest statement in the “Confidential to Editor” section, and submit your "Accept" recommendation.

Reviewer #1: (No Response)

Reviewer #2: All comments have been addressed

2. Is the manuscript technically sound, and do the data support the conclusions?

Reviewer #1: Yes

Reviewer #2: Yes

3. Has the statistical analysis been performed appropriately and rigorously? 

Reviewer #1: Yes

Reviewer #2: Yes

4. Have the authors made all data underlying the findings in their manuscript fully available?

Reviewer #1: Yes

Reviewer #2: Yes

5. Is the manuscript presented in an intelligible fashion and written in standard English?

Reviewer #1: Yes

Reviewer #2: Yes

6. Review Comments to the Author

Reviewer #1: Comments to the Author

I have reviewed the revised manuscript “Seismic performance of base-isolated structures for swimming pool reactors under Different foundation conditions” by Jie Zhao,Jianshan Wang,Jiehua Huang. I invite the authors to add some comments in the text regarding the following comments:

- Within Introduction and text, discuss that a numerical model is always affected by two families of uncertainties (i.e., aleatory and epistemic), as discussed in:

Gino, D. et al., Strain-based method for assessment of global resistance safety factors for NLNAs of reinforced concrete structures, Engineering Structures, 2024, 304, 117625.

Miceli, E. et al., Approaches to estimate global safety factors for reliability assessment of RC structures using non-linear numerical analyses, Engineering Structures, 2024, 311, 118193.

In detail, comment how the both uncertainties can influence the results and which one can have a dominant role.

- Discuss if the records can be considered frequent or no-frequent ground motions as well as their PGA/PGV ratios, as described in:

Castaldo, P., Miceli, E. Optimal single concave sliding device properties for isolated multi-span continuous deck bridges depending on the ground motion characteristics, Soil Dynamics and Earthquake Engineering, 2023, 173, 108128.

In detail, provide the value of the ratio “PGA/PGV” discussing if it is a high or medium or low value.

Reviewer #2: The revisions made by the authors in response to my comments are satisfactory, and the paper is recommended for publication.

7. PLOS authors have the option to publish the peer review history of their article (what does this mean?). If published, this will include your full peer review and any attached files.

Reviewer #1: **Yes: **Paolo Castaldo

Reviewer #2: No

---

## [Author Response · Author response to Decision Letter 1]

9 Sep 2024

Title: Seismic performance of base-isolated structures for swimming pool reactors under Different foundation conditions

Authors: Jie Zhao, Jianshan Wang, Jiehua Huang

Paper No.: PONE-D-24-23674R1

Dear Editor,

The authors would like to thank the reviewers for their valuable comments and suggestions. The reviewers' opinions have been considered in the revised manuscript, and four additional references have been added according to their suggestions. The authors believe that these recommendations have significantly improved the quality of the paper. Below are our responses to the questions and concerns raised by the reviewers.

Reply to Reviewer 1’s Comments:

1. Within Introduction and text, discuss that a numerical model is always affected by two families of uncertainties (i.e., aleatory and epistemic), as discussed in:

Gino, D. et al., Strain-based method for assessment of global resistance safety factors for NLNAs of reinforced concrete structures, Engineering Structures, 2024, 304, 117625.

Miceli, E. et al., Approaches to estimate global safety factors for reliability assessment of RC structures using non-linear numerical analyses, Engineering Structures, 2024, 311, 118193.

In detail, comment how the both uncertainties can influence the results and which one can have a dominant role.

Reply：Thank you for the expert's opinion. In numerical model analysis, there are always two types of uncertainties: aleatory uncertainty (i.e., materials, geometric properties and actions) and epistemic uncertainty (i.e., modeling). Aleatory uncertainty arises from the inherent randomness in data or models. For example, there may be measurement noise in the experimental data, or there may be random disturbances in the simulation. Epistemic uncertainty arises from incomplete knowledge of the model or system, including simplification, approximation, or inaccuracy of assumptions. For example, simplification of physical models, roughness of computational grids, etc. By improving the model, increasing computational accuracy, and reducing assumptions, cognitive uncertainty can be reduced. This article is dominated by epistemic uncertainty, and the correctness of viscous-spring boundary is verified through numerical examples to reduce epistemic uncertainty. Within Introduction and text, discuss that a numerical model is always affected by two families of uncertainties (i.e., aleatory and epistemic), and cited four relevant literature, including two articles recommended by the reviewers. The opinions of experts are very forward-looking, and I will focus on studying aleatory and epistemic uncertainty in future research

2. Discuss if the records can be considered frequent or no-frequent ground motions as well as their PGA/PGV ratios, as described in:

Castaldo, P., Miceli, E. Optimal single concave sliding device properties for isolated multi-span continuous deck bridges depending on the ground motion characteristics, Soil Dynamics and Earthquake Engineering, 2023, 173, 108128.

In detail, provide the value of the ratio “PGA/PGV” discussing if it is a high or medium or low value.

Reply：The expert's opinion is very good. When using PGA/PGV for long structures such as long-span cable-stayed bridges, there will be significant changes in the displacement response of the structure. The structure of this article is a pool type reactor, not a long structure. According to the "Code for Seismic Design of Buildings" and "Seismic Design Standards for Nuclear Power Plants", the seismic parameters used are mostly the peak ground acceleration (PGA) and acceleration response spectrum. This article records seismic waves that cannot be considered frequent or infrequent. The seismic waves used are an improved version of the 1.60 Seismic Design Response Spectrum developed by the US Nuclear Regulatory Commission in 1973, which is used for seismic response analysis based on the seismic safety review standards for nuclear power plants. The corresponding response spectrum is a broadband spectrum that covers various damping conditions and is conservatively designed to typically cover specific or expected locations in the country, thus well describing the seismic capacity of nuclear facilities. The total duration is 28 seconds, with a time step of 0.01 seconds. The PGA value is 0.3g in the X and Z directions and 0.2g in the Y direction. There is a detailed explanation on line 287. Thank you for your interesting feedback. In future research, we will consider the impact of seismic parameters using PGA/PGV on the structure.

Please feel free contact us if further revision is needed.

Thank you very much for your attention.

Best regards,

Dr. Zhao

School of Architectural Engineering

Dalian University

Dalian 116622

P. R. China

E-mail: zhaojie_gd@163.com

---

## [Decision Letter · Decision Letter 2]

17 Sep 2024

Seismic performance of base-isolated structures for swimming pool reactors under Different foundation conditions

PONE-D-24-23674R2

Dear Dr. Jie Zhao

We’re pleased to inform you that your manuscript has been judged scientifically suitable for publication and will be formally accepted for publication once it meets all outstanding technical requirements.

Kind regards,

Mohammadreza Vafaei, Ph.D.

Academic Editor

PLOS ONE

Additional Editor Comments (optional):

Reviewers' comments:

Reviewer's Responses to Questions

**Comments to the Author**

1. If the authors have adequately addressed your comments raised in a previous round of review and you feel that this manuscript is now acceptable for publication, you may indicate that here to bypass the “Comments to the Author” section, enter your conflict of interest statement in the “Confidential to Editor” section, and submit your "Accept" recommendation.

Reviewer #1: All comments have been addressed

2. Is the manuscript technically sound, and do the data support the conclusions?

Reviewer #1: Yes

3. Has the statistical analysis been performed appropriately and rigorously? 

Reviewer #1: Yes

4. Have the authors made all data underlying the findings in their manuscript fully available?

Reviewer #1: Yes

5. Is the manuscript presented in an intelligible fashion and written in standard English?

Reviewer #1: Yes

6. Review Comments to the Author

Reviewer #1: Comments to the Author

I have reviewed the revised manuscript “Seismic performance of base-isolated structures for swimming pool reactors under Different foundation conditions” by Jie Zhao,Jianshan Wang,Jiehua Huang. The revised text can be published.

7. PLOS authors have the option to publish the peer review history of their article (what does this mean?). If published, this will include your full peer review and any attached files.

Reviewer #1: **Yes: **Paolo Castaldo

---

## [Editor Report · Acceptance letter]

9 Oct 2024

PONE-D-24-23674R2 

PLOS ONE

Dear Dr. Zhao, 

I'm pleased to inform you that your manuscript has been deemed suitable for publication in PLOS ONE. Congratulations! Your manuscript is now being handed over to our production team.

Kind regards, 

on behalf of

Dr. Mohammadreza Vafaei 

Academic Editor

PLOS ONE